# Uncovering interpretable potential confounders in electronic medical records

Jiaming Zeng [1✉], Michael F. Gensheimer[2], Daniel L. Rubin[3], Susan Athey [4] & Ross D. Shachter [1]

Randomized clinical trials (RCT) are the gold standard for informing treatment decisions. Observational studies are often plagued by selection bias, and expert-selected covariates may insufficiently adjust for confounding. We explore how unstructured clinical text can be used to reduce selection bias and improve medical practice. We develop a framework based on natural language processing to uncover interpretable potential confounders from text. We validate our method by comparing the estimated hazard ratio (HR) with and without the confounders against established RCTs. We apply our method to four cohorts built from localized prostate and lung cancer datasets from the Stanford Cancer Institute and show that our method shifts the HR estimate towards the RCT results. The uncovered terms can also be interpreted by oncologists for clinical insights. We present this proof-of-concept study to enable more credible causal inference using observational data, uncover meaningful insights from clinical text, and inform high-stakes medical decisions.

[1] Department of Management Science and Engineering, Stanford University, Stanford, CA 94305, USA. [2] Department of Radiation Oncology, Stanford School of Medicine, Stanford, CA 94305, USA. [3] Department of Biomedical Data Science, Radiology, and Medicine, Stanford School of Medicine, Stanford, CA 94305, USA. [4] Graduate School of Business, Stanford University, Stanford, CA 94305, USA. ✉email: jiaming@alumni.stanford.edu

As the number of highly targeted cancer treatments increases, it is increasingly difficult for oncologists to decide on optimal treatment practices. In recent years, medicine has seen the reversal of 146 standard medical practices[1], and many unanswered questions remain on treatment decisions in oncology. The gold standard for assessing treatment effects is randomized clinical trials (RCT). However, RCTs can be very expensive, time-consuming, and limited by the lack of external validity[2,3]. Hence, there has been a growing interest in using observational data to compare and evaluate the effectiveness of clinical interventions, also known as comparative effectiveness research (CER)[2].

Many studies have used large-scale observational registries such as the Surveillance, Epidemiology, and Ends Results (SEER) and National Cancer Data Base (NCDB) to perform CER. However, such studies may be unreliable due to the systemic bias present in observational data and the presence of unmeasured confounders[1,2,4]. Moreover, population-based CERs in oncology often also face small data challenges. Electronic medical records (EMRs) are another source of rich observational information on patient demographics and past medical history. We hypothesize that the more detailed unstructured data present in EMRs can be harnessed to reduce confounding compared to prior CER studies.

In the past decade, there has been a growing interest in using observational data for clinical decision-making and causal inference in oncology[2]. However, such studies are often unreliable, and many observational studies have been refuted by RCTs soon after[2,4]. For example, Yeh et al.[5] performed a comparison of surgery vs. radiotherapy for oropharynx cancer and suggested that surgery may be superior to radiation for quality of life outcomes. A few years later, this claim was refuted by an RCT study by Nichols et al.[6], which showed that radiation is in fact superior to surgery in terms of 1-year quality of life scores. A similar example is seen with prostate cancer. In 2016, Wallis et al.[7] showed through population-based studies that surgery is superior to radiation for early-stage prostate cancer for overall and prostate cancer-specific survival; a few months later, the finding was refuted by Hamdy et al.[8], which showed that surgery and radiation are equivalent in terms of overall and prostate cancer-specific survival. Many other studies have shown the fallibility of population CERs that rely on expert-curated features to draw conclusions about treatment effects[2,9].

Beyond clinical studies, there is a relatively large literature on performing causal inference from observational data. Various papers have explored how to correct for bias when evaluating average treatment effect (ATE) from observational studies with propensity score matching or weighting[10,11]; see ref. [12] for a review. There is also a growing amount of literature that adapts machine-learning models, such as random forest or regularized regression, for doubly robust ATE estimation in high-dimensional settings[13–16]. However, most of the methods do not include unstructured data.

Recent literature has shown the usefulness of conditioning on textual data to adjust for confounding[17–20]. Roberts et al.[18] propose text matching to employ textual data for causal inference. Mozer et al.[17] apply text matching to patient charts texts for a medical procedure evaluation; however, they focus on continuous outcomes and rely mostly on expert-curated terms from the clinical text. Veitch et al.[19] is another work that employs unstructured data for causal inference; however, they rely on black-box models that are not interpretable. Moreover, many existing causal inference methods are developed for continuous outcomes and do not transfer easily to the time-to-event outcomes for survival analysis used in oncology. Of the ones that perform causal inference on time-to-event outcomes for medical applications[21,22], we did not find any that include unstructured

data in a systematic way. Austin[22] presents methods for using propensity scores to reduce bias in observational studies with time-to-event outcomes. Our study leverages some of the ideas and methods in this literature to develop our approach for identifying and evaluating the potential confounders from the unstructured clinical notes. Keith et al.[20] present a review of the literature on using textual data to adjust for confounding. Our paper contributes to this literature by addressing obstacles in using NLP methods to remove confounding.

There is also a growing literature that seeks to better employ EMRs for clinical tasks. Existing work has employed structured EMR data and unstructured clinical notes for survival prediction and analysis[23,24], clinical risk prediction[25], and prediction of multiple medical events[26]. However, most current work involving EMRs focuses on prediction tasks. In studies that include unstructured notes, most use deep learning to produce context-rich embedding representations of words or documents[23,26]. While these representations are highly accurate for prediction tasks, they are often black-box and very difficult to interpret for causal insights. Our approach differs in that we use simple NLP techniques (e.g., entity identification, bag-of-words) to generate matrix representations that can be easily mapped to specific words and phrases. This increases the interpretability of our method and allows us to explain our confounders to clinicians.

We study how EMRs, especially clinical text, can be used to reduce selection bias in observational CER studies and better inform treatment decisions in oncology. A confounder is a variable that is associated with both treatment assignment and the potential outcomes a subject would have under different treatment regimes. In the presence of confounders, the correlation between treatment assignment and outcomes cannot be interpreted as causal. One way that confounding may arise is when patients are selected for a treatment group on the basis of the severity of their illness. In such a case, failing to adjust for patient severity can lead to selection bias when attempting to estimate causal effects. For example, surgery tends to be performed on younger or healthier patients; certain doctors or institutions may prefer one treatment over another, and this creates confounding if those doctors or institutions treat patients with systematically different severity. Studies based on a small set of covariates tend not to capture the important confounders and result in biased estimates[5,7]. Observational studies are more reliable when we can better control for these confounders. While structured EMR data, such as billing codes, can be used to encode expert-curated patient characteristics, studies suggest that administrative claims data may contain errors[27,28] and expert-curated covariates may not capture all potential confounding[7,29]. EMR clinical text is a potential source of additional information about factors that might relate to both treatment assignment and prognosis.

We propose an automated approach using natural language processing (NLP) to uncover interpretable potential confounders from the EMRs for treatment decisions. For high-stake settings such as cancer treatment decisions, it is important to design models that are interpretable for trust and understanding[30]. NLP can be used to process the unstructured clinical notes and create covariates that supplement the traditional covariates identified through expert opinion. We then augment our dataset with covariates that impact both treatment assignment and patient outcomes, where attempting to estimate causal effects while omitting such variables leads to biased estimates[15,31]. Finally, we use methods designed to estimate causal effects in observational studies with observed confounders to estimate treatment effects in our augmented dataset. We show that controlling for these confounders appears to reduce selection bias when compared against the results from established RCTs and clinical judgment.

We apply our method to localized prostate and lung cancer patients. Based on cohorts from established RCTs, we built four treatment groups for comparison. We uncovered interpretable potential confounders from clinical text and validated the potential confounders against the results from the RCTs. Simple NLP techniques (e.g., lemmatization, entity identification) were used to construct a bag-of-words representation of the frequently occurring terms. A Lasso model[32] was then used to select the terms that are predictive of both the treatment and survival outcome as potential confounders. Finally, we validated our method by comparing the hazard ratio (HR) from survival analysis with and without the confounders.

Our main contribution is presenting a framework for uncovering interpretable potential confounders from clinical text. Existing work in observational causal inference rarely employs unstructured data[11,21,22], and most NLP studies on clinical text focus on prediction or classification settings[23,26,28]. Our paper differs from existing studies by employing NLP for causal analysis; we use NLP methods to predict the treatment and survival outcome, and then employ a causal framework to combine the two models for uncovering potential confounders. We are the first to uncover interpretable potential confounders from clinical notes for causal analysis on cancer therapies, and one of the few works that combine NLP and causal inference in a time-to-event setting. Our method allows researchers to extract and control for confounders that are not typically available. While we present our work as a proof-of-concept study, this appears to be a useful step for future observational CER studies to help reduce selection bias unique to that dataset. The research presented can help unlock the potential of clinical notes to help clinicians understand the current clinical practice and support future medical decisions. We also outline several limitations that need to be overcome for use in practice in "Discussion".

Our study advances both the clinical and causal inference literature by using NLP to perform causal inference on clinical text in time-to-event settings. We hope this will inform clinical practice and improve patient outcomes.

## Results

We apply our methods to localized prostate and stage I non-small cell lung cancer (NSCLC) patients and compare the results against established RCTs. We select these diseases due to data availability and having established clinical RCTs for validation. After filtering and assignment, we include 1822 patients for prostate cancer, with 988 surgery patients, 385 radiation patients, and 449 active monitoring patients; the average follow-up time is 4.11 years. For stage I NSCLC, we include 749 patients, with 492 surgery patients and 257 radiation patients; the average follow-up time is 4.96 years. The patient characteristic descriptions of the prostate cancer cohort are shown in Table 1 and the NSCLC cohort are shown in Table 2. Please see "Dataset" for more details on the patient selection process.

We use the findings from established RCTs and clinical judgment as a benchmark for evaluating our results. For localized prostate cancer, Hamdy et al.[8] compared active monitoring, radical prostatectomy, and external-beam radiotherapy. A total of 1643 patients were included in the study, with 553 men assigned to surgery, 545 men assigned to radiotherapy, and 545 men to active monitoring. They observed no significant difference among the groups for prostate cancer or all-cause mortality ($P = 0.48$ and $P = 0.87$ respectively). Similarly, a recent study showed that the difference in treatment effects for surgery vs. radiation observed from observational studies is entirely due to treatment selection bias[29]. For stage I NSCLC, the Chang et al.[33] study is a pooled study comparing stereotactic ablative radiotherapy

**Table 1 Characteristics of the localized prostate cancer patients.**

| Features | Treatment groups | | |
| --- | --- | --- | --- |
| | Surgery (*n* = 988) | Radiation (*n* = 385) | Monitoring (*n* = 449) |
| Age, mean (std) | 64.04 (7.8) | 70.18 (7.6) | 66.22 (8.2) |
| Race, no. (%) | | | |
| White | 709 (71.8%) | 221 (57.4%) | 292 (65.0%) |
| Black | 32 (3.2%) | 15 (3.9%) | 14 (3.1%) |
| Asian | 94 (9.5%) | 41 (10.6%) | 42 (9.4%) |
| Unknown | 153 (15.4%) | 108 (28.1%) | 101 (22.5%) |
| Ethnicity, no. (%) | | | |
| Hispanic | 71 (7.1%) | 20 (5.2%) | 23 (5.1%) |
| Non-Hispanic | 890 (90.1%) | 348 (90.4%) | 393 (87.5%) |
| Unknown | 27 (2.7%) | 17 (4.4%)) | 33 (7.3%) |
| Clinical stage, no. (%) | | | |
| Stage I | 219 (22.2%) | 36 (9.4%) | 227 (50.6%) |
| Stage II | 750 (75.9%) | 289 (75.1%) | 217 (48.3%) |
| Stage III | 12 (1.2%) | 38 (9.9%) | 3 (0.7%) |
| Stage IV | 7 (0.7%) | 22 (5.7%) | 2 (0.4%) |
| Tumor grade, no. (%) | | | |
| Grade 1 | 66 (66.8%) | 33 (8.6%) | 157 ((35.0%) |
| Grade 2 | 429 (43.4%) | 132 (34.3%) | 205 (45.7%) |
| Grade 3 | 474 (48.0%) | 208 (54.0%) | 62 (13.8%) |
| Grade 4 | 3 (0.3%) | 2 (0.5%) | 0 (0%) |
| Unknown | 16 (1.6%) | 10 (2.6%) | 25 (5.6%) |
| No. notes/patient, mean (std) | 24.96 (44.4) | 53.93 (105.7) | 54.48 (93.4) |
| Days of survival, mean (std) | 1564.90 (979.4) | 1424.76 (1,031.6) | 1403.72 (921.2) |
| Death, no. (%) | 70 (7.1%) | 19 (4.9%) | 17 (3.8%) |

Diagnosis year: 2008–2017; avg. follow-up: 4.11 years.

**Table 2 Characteristics of the stage I lung cancer patients.**

| Features | Treatment groups | |
| --- | --- | --- |
| | Surgery (*n* = 484) | Radiation (*n* = 224) |
| Age, mean (std) | 68.05 (10.7) | 74.60 (9.1) |
| Gender, no. (%) | | |
| Female | 299 (62.0%) | 87 (41.2%) |
| Male | 185 (38.0%) | 137 (58.8%) |
| Race, no. (%) | | |
| White | 293 (60.8%) | 152 (66.5%) |
| Black | 12 (2.2%) | 5 (3.5%) |
| Asian and Pacific islander | 99 (20.1%) | 18 (9.7%) |
| Unknown | 80 (16.9%) | 49 (20.2%) |
| Ethnicity, no. (%) | | |
| Hispanic | 23 (4.9%) | 10 (3.9%) |
| Non-hispanic | 411 (84.3%) | 178 (81.7%) |
| Unknown | 50 (10.8%) | 36 (14.4%) |
| No. notes/patient, mean (std) | 57.49 (101.2) | 57.73 (134.9) |
| Days of survival, mean (std) | 2060.13 (1,207.5) | 1350.29 (914.1) |
| Death, no. (%) | 120 (24.8%) | 126 (53.3%) |

Diagnosis year: 2000–2017; avg. follow-up: 4.96 years.

(SABR) to surgery. A total of 58 patients were included, with 31 patients assigned to SABR and 27 to surgery. The study observed that SABR had slightly better overall survival than surgery ($P = 0.037$), but claims to be consistent with the clinical judgment that surgery is equipoise to radiation.

Following the design of Hamdy et al.[8] and Chang et al.[33], we evaluate our results for the following four treatment groups for an outcome of all-cause mortality:

- Surgery vs. radiation for prostate cancer
- Surgery vs. monitoring for prostate cancer

- Radiation vs. monitoring for prostate cancer
- Surgery vs. radiation for stage I NSCLC

We do not analyze other treatment groups for lung cancer due to patient count constraints.

Our approach identifies covariates that are likely potential confounders in this particular dataset from the high-dimensional and high-noise EMR data. These covariates are interpretable as they are represented by structured data or words from a bag-of-words matrix. To evaluate the effectiveness of the potential confounders selected in the model, we use these potential confounders to perform survival analysis for the treatment groups for prostate and stage I NSCLC. We compare the results of various methods for time-to-event analysis in terms of HR. Although we cannot know what the true HR is, we suggest that using medical notes improves on the traditional covariates. We compare our results against existing RCTs to evaluate how the confounders we have uncovered can help correct selection bias. The overall workflow is shown in Fig. 1. Supplement 1 details the covariates extracted from the structured data.

**Potential confounders**. We show that our methods uncover terms that are predictive of both the treatment and survival outcome. Hence, these are potential confounders that should be controlled for in observational CERs to reduce selection bias. Please see Supplement 3 for a discussion on the structures of potential confounding our method can capture.

We select the intersection covariates from our treatment and outcome prediction models as the potential confounders. We base this idea on the selection of union variables to reduce confounding when performing causal inference on observational data in the case of continuous outcomes[15]. However, in survival analysis, it is recommended that the covariates analyzed be constrained by the statistical 1 in 10/20 rule of thumb with respect to the event count[34,35]. In our high-dimensional setting, the union of covariates that are predictive of treatment and outcome yields too many potential confounders relative to the sample size. Hence, we use intersect as a heuristic to focus on the most important confounders.

In Fig. 2, we illustrate the unpenalized coefficients of covariates from two models, the treatment assignment model, and the survival outcome model. For each covariate, the $x$ axis plots the coefficient from the treatment prediction model while the $y$ axis plots the coefficient from the survival outcome model. Each covariate is labeled by the text next to it. The intersection covariates, intersect, are shown in blue; these are the covariates that have strong effects in both models. For the structured covariates, we illustrate in black the coefficients for the covariates that were not selected; these coefficients are closer to at least one of the axes in the figure. We do not illustrate the coefficients for unstructured covariates that are not selected, as there are a large number of these covariates. The axes are labeled to indicate which treatment the coefficient predicts and whether the coefficient is indicative of a good or bad survival prognosis. For example, in the treatment model, patients with a high bladder word occurrence have a higher likelihood of receiving surgery; in the outcome model, patients with a high bladder occurrence have a lower likelihood of survival.

In Supplement 5, we show the $R^2$ correlation among all the selected covariates for each treatment group.

**Evaluation of potential confounders**. We evaluate these potential confounders by comparing the results on three covariate combinations:

- Structured: Using only the structured covariates. We use this as a baseline because these are covariates that are typically used in retrospective oncology studies and are readily available in the structured data[7].
- Intersect: Using only the intersection covariates identified as confounders.
- Struct+intersect: Using the union of the structured and intersection variables.

We then perform survival analysis using univariate Cox proportional hazard models (Cox-PH) with propensity score matching (**matching**), univariate Cox-PH model with inverse propensity score weighting (**IPTW**), and multivariate Cox-PH model with inverse propensity score weighting (**multi.coxph**). We hypothesize that struct+intersect will perform the best by including both the structured and unstructured data. In Fig. 3, we show the hazard ratio (HR) of the effect of treatment for each study cohort when the selected covariates are included in the analysis. An HR below 1 indicates that patients with the second treatment are more likely to survive than those with the first treatments. An HR above 1 indicates the opposite, and an HR equal to 1 indicates that the two treatments are equipoise. For each HR estimate, we also show the 95% confidence interval (CI). Please see "Uncover and evaluate confounders" for more details on the methods.

We observe that with the additional covariates, we are able to shift the estimate of the HR toward the direction of the RCT for an outcome of all-cause mortality. We also compare the covariate-specific HR of each of the selected covariates in terms of univariate and multivariate Cox-PH analysis for an all-cause mortality outcome in Tables 3–6.

In Fig. 3a and Table 3, we show the results with surgery vs. radiation for prostate cancer. The RCT reports no significant difference between surgery vs. radiation for localized prostate cancer[8]. With structured, we observe a significant effect that radiation is superior to surgery, a result that disagrees with most retrospective studies[7]. Each center can have different patient populations and treatment patterns that shift the only structured adjusted survival rates. For instance, at our center we have a busy high-dose-rate brachytherapy program which is an attractive option for fit patients with few comorbidities who might otherwise receive surgery. This would be expected to bias the survival outcomes in favor of radiation, as observed in our study. We seek to uncover potential confounders from the text that can reduce bias when performing retrospective studies, whichever way the bias lies. After adjustment with the uncovered confounders, we observe a significant shift in the HR toward equipoise with the additional identified confounders for intersection and struct +intersect. For structured, we observe an HR of 2.51 with 95% CI (2.39–4.55) and $P$ value of 0.002 with **multi.coxph**. For struct +intersect, we estimate an HR of 1.54 with 95% CI (0.78–3.03) and $P$ value of 0.214 with **multi.coxph**. We shift the HR point estimate by 0.97, or 38.6%, toward equipoise.

In Fig. 3b and Table 4, we show the results of surgery vs. active monitoring for prostate cancer. Hamdy et al.[8], the RCT, reports the HR for surgery vs. active monitoring as 0.93 with 95% CI (0.65, 1.35) and $P$ value of 0.92. With structured, we again have a significant effect that active monitoring is superior to surgery; this disagrees with most retrospective studies[7] and Hamdy et al.[8]. We again observe a significant shift in the HR toward equipoise with the additional identified confounders. For structured, we observe an HR of 2.71 with 95% CI (1.55–4.75) and $P$ value < 0.001 with **multi.coxph**. For struct+intersect, we estimate an HR of 1.10 with 95% CI (0.55–2.21) and $P$ value of 0.781 with **multi.coxph**. We shift the HR point estimate by 1.61, or 59.1%, toward equipoise.

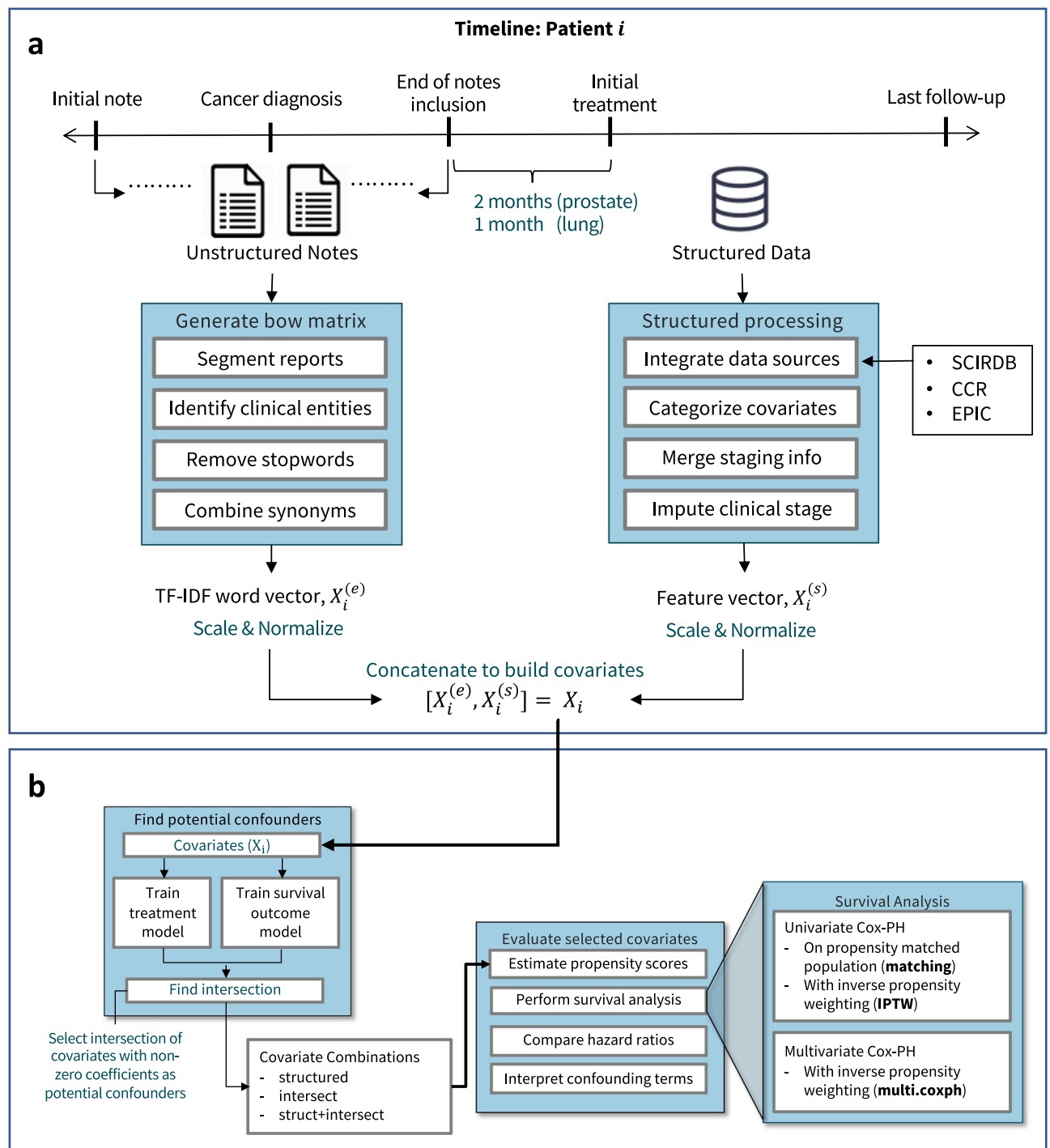

**Fig. 1 Pictorial overview for uncovering potential confounders. a** shows the data processing done for each patient. We preprocess and concatenate the structured and unstructured covariates before applying our method. For the data sources, we compile data from the Stanford Cancer Institute Research Database (SCIRDB), the California Cancer Registry (CCR), and the Epic System. We present the timeline for patient $i$ with both structured ($X_i^{(s)}$) and unstructured ($X_i^{(e)}$) features $X_i$. **b** shows the workflow for identifying how potential confounders affect survival analysis for each treatment group. We uncover covariates that are predictive of both the treatment and outcome as potential confounders. We then perform survival analysis on different combinations of the selected covariates.

In Fig. 3c and Table 5, we show the results of radiation vs. active monitoring for prostate cancer. We do not see as significant a shift with radiation vs. active monitoring. Hamdy et al.[8] record the HR for radiation vs. active monitoring as 0.94 with 95% CI of (0.65, 1.36) and $P$ value of 0.92. We observe that **matching** estimated the HR closest to the RCT results when compared against **IPTW** and **multi.coxph**. All results with intersect and struct+intersect shift the HR estimate slightly toward equipoise, with the most shift of 0.32, or 71.1%, by intersect and **IPTW**; this is closely followed by a shift of 0.20, or 45.5%, with intersect and **multi.coxph**. While the adjusted results are not as close to the RCT results as compared to Fig. 3a and b, the HR estimate is all shifted toward the RCT results in terms of bias reduction for each of the data and method combination. We suspect the less

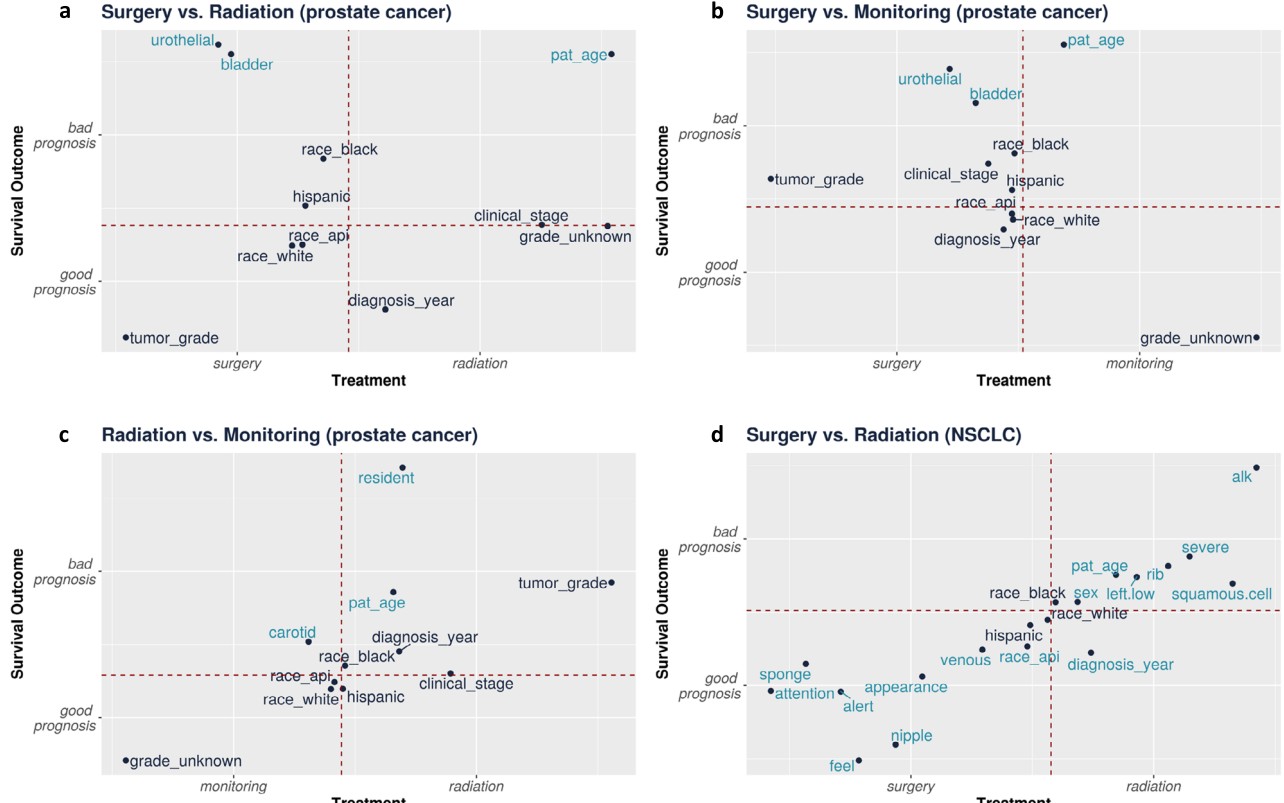

**Fig. 2 For each treatment group, we show the unpenalized coefficients for the struct+intersect covariates.** Blue text indicates the intersect covariates that have been selected as potential confounders by our method from the text; the prefix text: has been omitted. Black text indicates the structured covariates that have not been selected; the prefix "struct:" has been omitted. The covariate patient_age has been shorthanded as pat_age. For the treatment model, these are the coefficients to a linear model. For the survival outcome model, these are the $\beta$ for the Cox-PH model. The dotted lines are the axis, denote a coefficient value of 0. We plot the coefficients from the survival and treatment regression models in each of the figure panels: **a** surgery vs. radiation for prostate cancer; **b** surgery vs. monitoring for prostate cancer; **c** surgery vs. monitoring for prostate cancer; **d** surgery vs. radiation for NSCLC.

significant shift may be due to the smaller dataset available for radiation vs. active monitoring or the confounding not being observable within the text.

In Fig. 3d and Table 6, we show the results with surgery vs. radiation for stage I NSCLC. With structure, we observe a significant effect that surgery is superior to radiation. The results from Chang et al.[33] and clinical judgment tells us that surgery and radiation should be about equipoise for stage I NSCLC. The shift is not as significant as with prostate cancer, but we also note that the established clinical standard for lung cancer is not as well studied. We observe a more significant shift with **multi.coxph**, with an average shift of 0.15 or 38.5%. We observe an average shift of 0.06, or 15.4%, with **matching** and an average shift of 0.02, or 5.1% with **IPTW**. For structured, we observe an HR of 0.39 with 95% CI (0.30–0.51) and *P* value < 0.001 with **multi.-coxph**. For struct+intersect, we estimate an HR of 0.54 with 95% CI (0.40–0.53) and *P* value < 0.001 with **multi.coxph**. We shift the HR point estimate by 0.15, or 38.5%, toward equipoise. While the adjusted results are not as close to the RCT results as compared to Fig. 3a and b, the HR estimates are all shifted towards equipoise in terms of bias reduction for each combination. We suspect the less significant shift is again due to the even smaller data size of stage I NSCLC. The doubly robust method of **multi.coxph** seem to perform better under these settings.

Overall, our methods uncover several potential confounders that can reduce selection bias in observational data. Although our method cannot uncover all potential confounders, we are able to

uncover confounders that are not usually included in expert-selected covariates. Supplementary analysis of propensity scores and covariate balance plots for each analysis is seen in Supplement 4.

**Potential confounder interpretation.** We show that the potential confounders we have uncovered are interpretable through clinical expertise. We examine the effect on survival for each selected covariate in term of univariate and multivariate survival analysis with a Cox-PH model. In univariate analysis, a single covariate is regressed on the survival outcome and describes the survival with respect to a single covariate. In multivariate analysis, all the selected covariates are regressed on the survival outcome and describe each covariate's effect on survival while adjusting for the impact of all selected covariates. For a particular variable, an HR below 1 indicates that the covariate is a positive predictor of survival, an HR above 1 indicates a negative predictor of survival, and an HR equal to 1 means that the variable does not seem to effect survival.

For surgery vs. radiation and surgery vs. active monitoring with prostate cancer, struct:patient_age, text:bladder, and text:urothelial are chosen as intersection covariates. Moreover, they are also shown to be significant through both univariate and multivariate covariate analysis in Tables 3 and 4.

Patient age is a known confounder in treatment decision and survival outcomes. Older patients are more likely to receive

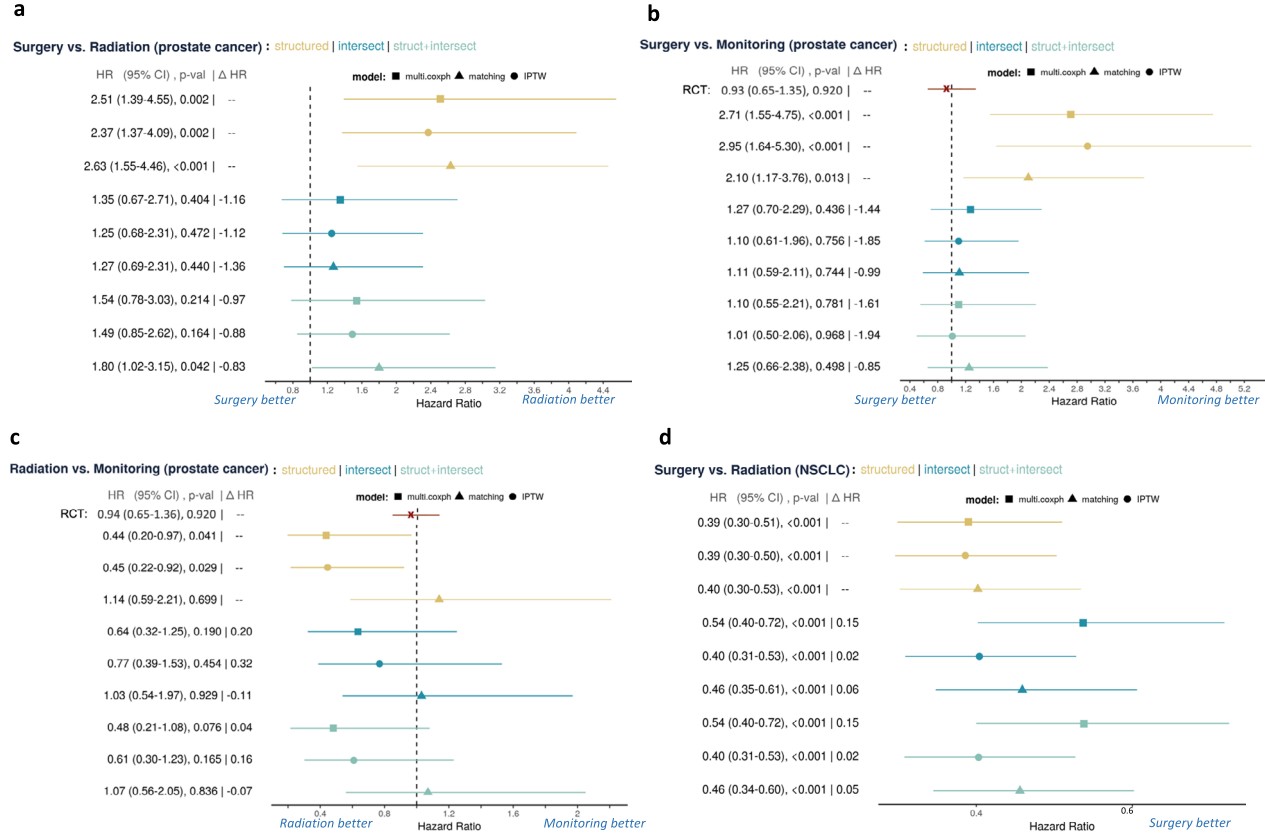

**Fig. 3 Forest plots of each of the comparison groups.** The left-hand label displays the mean HR, the 95% confidence interval (CI), the P value, and the Δ HR. The Δ HR measure is the difference of the current HR estimate and the baseline, structured, HR estimate. For surgery vs. active monitoring and radiation vs. active monitoring for prostate cancer, we have included the exact results from the RCT in red for comparison. For the remaining cohorts, clinical expertise suggests equipoise between the treatments. We see that the inclusion of our potential confounders shift the HR point estimate in the direction of the RCT and reduces the selection bias. The blue labels below each graph indicate which treatment is better in terms of HR comparison. We display the HR comparisons for each cohort as follows: **a** surgery vs. radiation for prostate cancer; **b** surgery vs. monitoring for prostate cancer; **c** surgery vs. monitoring for prostate cancer; **d** surgery vs. radiation for NSCLC. For (**d**), "Radiation better" is not displayed because the HR value is not shifted beyond 1.0, the direction of radiation better.

**Table 3 Univariate and multivariate covariate-specific HR for surgery vs. radiation for prostate cancer.**

| Covariates | Univariate analysis | | | Multivariate analysis | | |
| --- | --- | --- | --- | --- | --- | --- |
| | HR | 95% CI | P value | HR | 95% CI | P value |
| W.surgery | 1.27 | [0.77, 2.1] | 0.352 | 1.09 | [0.59, 2] | 0.777 |
| struct:patient_age* | 594.88 | [87, 4.1e+03] | <0.001 | 35.96 | [3.5, 3.7e+02] | 0.003 |
| struct:race_white | 0.92 | [0.44, 1.9] | 0.822 | 0.65 | [0.22, 1.9] | 0.439 |
| struct:race_api | 0.63 | [0.18, 2.2] | 0.467 | 0.67 | [0.14, 3.3] | 0.622 |
| struct:race_black | 1.63 | [0.33, 8.1] | 0.551 | 4.04 | [0.64, 25] | 0.137 |
| struct:hispanic | 0.85 | [0.2, 3.6] | 0.831 | 1.52 | [0.33, 7] | 0.593 |
| struct:clinical_stage | 0.30 | [0.042, 2.2] | 0.237 | 1.02 | [0.14, 7.4] | 0.987 |
| struct:tumor_grade | 0.05 | [0.0013, 2] | 0.111 | 0.10 | [0.00038, 24] | 0.406 |
| struct:grade_unknown | 0.55 | [0.028, 11] | 0.698 | 0.99 | [0.0029, 3.4e+02] | 0.996 |
| struct:diagnosis_year | 0.12 | [0.024, 0.57] | 0.008 | 0.17 | [0.025, 1.2] | 0.075 |
| text:bladder* | 207.51 | [79, 5.4e+02] | <0.001 | 35.95 | [9.3, 1.4e+02] | <0.001 |
| text:urothelial* | 1919.54 | [4.2e+02, 8.7e+03] | <0.001 | 44.07 | [4.4, 4.4e+02] | 0.001 |

HR hazard ratio, CI confidence interval, * intersection terms.
The * denotes intersection terms identified by our method. The lower block of covariates represents terms extracted from clinical notes. For each covariate, we show the effect size (HR), the 95% confidence interval (CI), and the statistical significance (P value) from a Wald statistics test.

radiation due to surgery risk. However, older patients also have higher mortality. In Fig. 2a–c, we observe that patients with higher struct:patient_age, i.e., older patients, are more likely to receive radiation and a bad prognosis.

We hypothesize that text:bladder and text:urothelial are identified because prostate cancer patients often have bladder symptom issues and can also have urothelial cancer. Most retrospective prostate cancer studies have not excluded patients

**Table 4 Univariate and multivariate covariate-specific HR for surgery vs. active monitoring for prostate cancer.**

| Covariates | Univariate analysis | | | Multivariate analysis | | |
|---|---|---|---|---|---|---|
| | HR | 95% CI | P value | HR | 95% CI | P value |
| W.surgery | 1.67 | [0.99, 2.8] | 0.057 | 1.02 | [0.55, 1.9] | 0.957 |
| struct:patient_age* | 3669.74 | [5.3e+02, 2.5e+04] | <0.001 | 143.94 | [11, 1.9e+03] | <0.001 |
| struct:race_white | 0.87 | [0.41, 1.8] | 0.709 | 0.68 | [0.23, 2] | 0.478 |
| struct:race_api | 0.59 | [0.15, 2.3] | 0.443 | 0.81 | [0.16, 4.1] | 0.799 |
| struct:race_black | 2.04 | [0.41, 10] | 0.384 | 5.16 | [0.82, 32] | 0.080 |
| struct:hispanic | 1.09 | [0.29, 4.1] | 0.898 | 1.68 | [0.41, 6.8] | 0.471 |
| struct:clinical_stage | 2.58 | [0.31, 21] | 0.378 | 3.75 | [0.35, 40] | 0.275 |
| struct:tumor_grade | 0.22 | [0.014, 3.7] | 0.296 | 2.37 | [0.0084, 6.6e+02] | 0.764 |
| struct:grade_unknown | 0.06 | [0.00094, 4.2] | 0.198 | 0.02 | [2.5e-05, 14] | 0.235 |
| struct:diagnosis_year | 0.12 | [0.027, 0.55] | 0.006 | 0.50 | [0.073, 3.4] | 0.483 |
| text:bladder* | 160.34 | [65, 3.9e+02] | <0.001 | 24.17 | [6.6, 89] | <0.001 |
| text:urothelial* | 2178.75 | [5e+02, 9.6e+03] | <0.001 | 68.15 | [7.4, 6.3e+02] | <0.001 |

HR hazard ratio, CI confidence interval, * intersection terms.
The * denotes intersection terms identified by our method. The lower block of covariates represents terms extracted from clinical notes. For each covariate, we show the effect size (HR), the 95% confidence interval (CI), and the statistical significance (P value) from a Wald statistics test.

**Table 5 Univariate and multivariate covariate-specific HR for radiation vs. active monitoring for prostate cancer.**

| Covariates | Univariate analysis | | | Multivariate analysis | | |
|---|---|---|---|---|---|---|
| | HR | 95% CI | P value | HR | 95% CI | P value |
| W.radiation | 1.22 | [0.63, 2.4] | 0.551 | 0.62 | [0.24, 1.6] | 0.316 |
| struct:patient_age* | 265.19 | [9.1, 7.8e+03] | 0.001 | 275.03 | [3.7, 2.1e+04] | 0.011 |
| struct:race_white | 0.43 | [0.15, 1.3] | 0.129 | 0.39 | [0.099, 1.5] | 0.170 |
| struct:race_api | 1.38 | [0.29, 6.7] | 0.687 | 0.62 | [0.09, 4.3] | 0.626 |
| struct:race_black | 1.69 | [0.18, 16] | 0.646 | 1.90 | [0.19, 19] | 0.582 |
| struct:hispanic | 0.38 | [0.014, 11] | 0.572 | 0.39 | [0.016, 9.7] | 0.567 |
| struct:clinical_stage | 2.42 | [0.2, 30] | 0.491 | 1.12 | [0.056, 22] | 0.939 |
| struct:tumor_grade | 0.89 | [0.034, 23] | 0.942 | 533.75 | [0.015, 1.9e+07] | 0.240 |
| struct:grade_unknown | 0.12 | [0.0017, 8.9] | 0.337 | 0.00 | [3e-07, 32] | 0.220 |
| struct:diagnosis_year | 0.69 | [0.053, 9.1] | 0.781 | 4.96 | [0.1, 2.4e+02] | 0.417 |
| text:carotid* | 44.60 | [4.1, 4.9e+02] | 0.002 | 9.63 | [2.1, 43] | 0.003 |
| text:resident* | 185839.25 | [80, 4.3e+08] | 0.002 | 1288062.91 | [1e+03, 1.6e+09] | <0.001 |

HR hazard ratio, CI confidence interval, * intersection terms.
The * denotes intersection terms identified by our method. The lower block of covariates represents terms extracted from clinical notes. For each covariate, we show the effect size (HR), the 95% confidence interval (CI), and the statistical significance (P value) from a Wald statistics test.

with early-stage bladder cancer[7]. Examples of text:bladder in the clinical notes are "he notes incomplete bladder emptying", "evidence of benign prostatic hyperplasia and chronic bladder outlet obstruction", and "diagnosis of bladder cancer". Examples of text:urothelial in the notes are "pathology showed high-grade urothelial carcinoma with muscle present and not definitively involved", "it was read as a high-grade urothelial cancer which involved the stroma of the prostate as well as the bladder". Patients with bladder cancer or bladder issues are more likely to get surgery than radiation. Radiation does not work well for bladder cancer. Patients with bladder problems may prefer surgery because radiation can irritate the bladder and cause urinary problems. However, these are also patients with higher mortality and more health issues. In Fig. 2a, b, we observe that text:bladder and text:urothelial are more common in patients who received surgery and had a bad prognosis.

Moreover, for this dataset, we note that the confounding appears to be observable. The bias of surgery being worse than radiation and monitoring is due to a group of patients who are diagnosed with prostate cancer through a resection for bladder cancer or other bladder issues. When a patient with bladder cancer has a cystoprostatectomy in which the bladder and prostate are both removed, a pathologist can sometimes find a prostate tumor in the pathology specimen. Bladder cancer

patients tend to be older, have more medical issues, and a higher mortality rate. The terms text:bladder and text:urothelial describe this group of patients. Our method can capture some characteristics of this group and use this to reduce selection bias.

For radiation vs. active monitoring, we do not observe confounders that present a significant shift in treatment HR in Table 5. It can be that the confounding here is not as easily observable or our method is unable to identify it. We can identify interesting potential confounders, such as text:resident. From Fig. 2c, we observe that text:resident is more common in patients who received radiation and had a bad prognosis. This term likely refers to both resident physicians and the patient being a resident of a long-term care facility or skilled nursing facility. Both uses of the term could reduce survival time: inpatients at teaching hospitals have much of their care delivered by resident physicians, and frequent inpatient stays or nursing facility residency could both indicate a sicker patient.

We repeat the same process for lung cancer. We examine Table 6 for the intersection covariates through univariate and multivariate analysis. We observe that some of the significant terms are struct:patient_age, struct:male, struct:race_api, struct:diagnosis_year, text:alk, text:left.low, and text:severe.

We note that age, gender, race, and diagnosis year are known confounders for treatment decision and outcome.

**Table 6 Univariate and multivariate covariate-specific HR for surgery vs. radiation for stage I NSCLC.**

| Covariates | Univariate analysis | | | Multivariate analysis | | |
|---|---|---|---|---|---|---|
| | HR | 95% CI | P value | HR | 95% CI | P value |
| W.surgery | 0.309 | [0.24, 0.4] | <0.001 | 0.545 | [0.41, 0.72] | <0.001 |
| struct:pat_age | 52.9 | [17, 1.6e+02] | <0.001 | 14.6 | [4.8, 44] | <0.001 |
| struct:male | 3.07 | [1.8, 5.1] | <0.001 | 1.92 | [1.1, 3.4] | 0.023 |
| struct:race_white | 0.842 | [0.53, 1.4] | 0.476 | 0.495 | [0.28, 0.87] | 0.015 |
| struct:race_api | 0.0557 | [0.019, 0.17] | <0.001 | 0.0674 | [0.021, 0.22] | <0.001 |
| struct:race_black | 2.03 | [0.62, 6.7] | 0.245 | 1.87 | [0.57, 6.1] | 0.298 |
| struct:hispanic | 0.664 | [0.19, 2.3] | 0.514 | 0.331 | [0.088, 1.2] | 0.101 |
| struct:diagnosis_year | 0.0166 | [0.007, 0.039] | <0.001 | 0.0421 | [0.014, 0.13] | <0.001 |
| text:alert | 9.46e-12 | [3.6e-16, 2.5e-07] | <0.001 | 0.00224 | [3e-08, 1.7e+02] | 0.287 |
| text:alk | 1.17e+04 | [38, 3.7e+06] | 0.001 | 4.67e+04 | [9.3e+02, 2.3e+06] | <0.001 |
| text:appearance | 5.46e-09 | [2.6e-12, 1.1e-05] | <0.001 | 0.00694 | [7.6e-06, 6.4] | 0.153 |
| text:attention | 1.03e-12 | [4.6e-20, 2.3e-05] | 0.001 | 0.00238 | [1.2e-09, 4.7e+03] | 0.414 |
| text:feel | 6.64e-10 | [5.7e-16, 0.00077] | 0.003 | 1.27e-05 | [1.2e-10, 1.4] | 0.057 |
| text:left.low | 29.1 | [5.2, 1.6e+02] | <0.001 | 12.5 | [2.1, 76] | 0.006 |
| text:nipple | 2.57e-09 | [1.6e-15, 0.0041] | 0.007 | 4.2e-05 | [1.5e-09, 1.2] | 0.054 |
| text:rib | 688 | [40, 1.2e+04] | <0.001 | 28.1 | [1.8, 4.4e+02] | 0.017 |
| text:severe | 601 | [56, 6.5e+03] | <0.001 | 58.3 | [9.4, 3.6e+02] | <0.001 |
| text:sponge | 2.04e-07 | [3.3e-11, 0.0013] | <0.001 | 0.0181 | [3.7e-05, 8.9] | 0.205 |
| text:squamous.cell | 94.8 | [16, 5.6e+02] | <0.001 | 7.61 | [1.1, 53] | 0.041 |
| text:venous | 0.0464 | [0.00089, 2.4] | 0.128 | 0.0523 | [0.00081, 3.4] | 0.165 |

*HR* hazard ratio, *CI* confidence interval, * intersection terms.
The * denotes intersection terms identified by our method. The lower block of covariates represents terms extracted from clinical notes. For each covariate, we show the effect size (HR), the 95% confidence interval (CI), and the statistical significance (P value) from a Wald statistics test.

The covariate text:alk points to the ALK mutation for NSCLC. About 5% of NSCLCs have a rearrangement in a gene called ALK; the ALK gene rearrangement produces an abnormal ALK protein that causes the cells to grow and spread. This change is often seen in non-smokers (or light smokers) who are younger and who have the adenocarcinoma subtype of NSCLC[36]. It's been observed that patients with the ALK mutation have worse disease-free survival, citing higher rates of recurrence and metastasis[36]. Alternatively, we hypothesize that text:alk is significant because the ALK mutation is mutually exclusive from the EGFR mutation[37]. The EGFR mutation is often present in asian patients and EGFR patients typically have better survival. Hence, the significance of text:alk can be related to the absence of the EGFR mutation. In Fig. 2d, we observe that text:alk is more common in patients who received radiation and had a bad prognosis.

The covariate text:left.low can point to NSCLC on the lower left node of the lung. Studies have observed that lung cancer on the lower lobe or lower left lobe has worse survival[38,39]. This can also be related to the absence of the EGFR mutation, since EGFR mutation occur less frequently in the lower lobe[38]. In Fig. 2d, we observe that text:left.low is also more common in patients who received radiation and had a bad prognosis.

The covariate text:nipple can indicate a history of breast cancer. Studies have shown that patients with a history of breast cancer are diagnosed with lower stages of NSCLC and show better prognosis when compared to women with first NSCLC, perhaps due to heightened surveillance compared to the general population[40]. In Fig. 2d, we observe that text:nipple is more common in patients who received surgery and had a good prognosis; both effects have also been observed in Milano et al.[40].

The covariate text:sponge can refer to sponges used for surgical preparations. The sponge is commonly used in surgery and can be an indication that the patient has some history of receiving surgery. Patients who receive surgery tend to be healthier and have better survival. In Fig. 2d, text:sponge is more common in patients who received surgery and had a good prognosis.

The covariates text:severe and text:rib could be pointing to a severe conditions related to lung and other problems that indicate poor overall health and performance status, which has been shown to be related to a patient's survival outcomes[41]. Examples of text:severe include phrases such as "severe pulmonary hypertension", "severe COPD", or "severe emphysema". Examples of text:rib include phrases such as "rib fractures" or "rib shadows". In Fig. 2d, we observe that both text:severe and text:rib are more common in patients who received radiation and had a bad prognosis. Similarly, we also observe other terms that could describe the type of lung cancer - such as text:squamous.cell—or overall health levels—textdstext:alert, text:attention.

Overall, we are able to uncover some potential confounders that are easy to interpret and capture useful clinical insights.

## Discussion

We have demonstrated how causal inference methods can be used to draw more reliable conclusions from population-based studies. Our paper shows that (1) clinical notes, or unstructured data, can be an important source for uncovering confounders, and (2) current clinical tools can be augmented with machine-learning methods to provide better decision support. Furthermore, our proof-of-concept framework can be easily adapted to use textual data to reduce selection bias in retrospective studies more generally.

Our framework can be used to improve clinical practice. Due to the simplicity of the machine-learning tools employed, it can be easily implemented as an additional step in the design of observational CER studies. Our results also show that the method is generalizable to different types of cancer and for various types of study cohort comparisons. With the continued digitization of clinical notes and the increasing access to EMRs, we recommend this as an essential step for any researcher seeking to draw clinical insights from observational data. The terms uncovered with our method can not only be used to improve observational CERs but

also be used to generate interpretable insights about current clinical practice. The uncovering of relevant information and subsequent insights can then be used to inform high-stakes medical decisions.

We believe that our work is the first to explore the potential of including unstructured clinical notes to reduce selection bias in oncology settings. We are also one of the first works to incorporate unstructured data into causal inference estimators and Cox-PH models. Although our method has been developed to address a specific problem in oncology and applied in the clinical setting, it can also be easily adapted for application in any observational study that seeks to incorporate unstructured text. We propose our method as an automated selection procedure that can be used to supplement expert opinion when uncovering potential confounders for a particular observational study population. There is much work to be done in using NLP and unstructured text for causal inference. Our work presents a simple and flexible way to generate interpretable causal insights from the text of any sort. Our method can also be applied to studies within and beyond medicine to extract important information from observational data to support decisions.

Our study also has several limitations. We begin by outlining potential areas for future work. First, we use simple NLP methods to process the clinical notes and extract the top 500 or 1000 features for variable selection. In the process, much information in the text nodes is discarded and the sequence of past medical events is not taken into account. We choose this setup due to the small sample size of oncology study cohorts, which makes it difficult to train more complicated models for textual processing. In theory, the more work that is placed into the clinical notes preprocessing and the higher quality of the features generated from these notes, the more informative the uncovered potential confounders will be. For future work, we hope to explore how other NLP techniques, such as topic modeling or clustering, can be used to build even higher-quality features from unstructured text. There are also an increasing number of deep learning models that can be used to identify interpretable insights[26]. We are interested in how these deep learning methods can be applied to generate causal insights on another study population with a larger sample size. We are also interested in developing ways to better address ambiguity in the notes (e.g., "it is unclear if the patient has chest pain").

Second, we rely on the proportional hazard assumption for our Cox-PH models. In cases of many covariates, the assumption may be violated. We feel the simplicity and interpretability of the model outweigh the performance improvement resulting from increased complexity. For EMR datasets with many covariates, the assumption is often used and does not seem to present a practical issue[24]. Future work could explore alternative models that do not rely on the assumption[42].

Third, more work can be done to mitigate immortal-time bias in our HR estimates. We discuss our approach in "Study cohort". An alternative method to address this problem would be to use a time-dependent Cox-PH model[43].

Fourth, we focus on the comparison of methods that can be applied in a time-to-event setting, and leave out more novel methods that are developed for continuous settings of ATE estimation. It would be interesting to explore how these methods can be extended and applied to a time-to-event setting.

Fifth, our approach of selecting intersection covariates is an empirical approach designed for uncovering the most valuable potential confounders. As a result, we filtered out most of the features and only focused on a few confounders. While our approach works well empirically in this study, future work involves developing more sensitive and statistically grounded methods for identifying potential confounders.

Sixth, our work is constrained to localized prostate and lung patients at the Stanford Hospital and state cancer registry data. It would strengthen the validity of our methods if experiments can be performed on large multiinstitutional registries for cancer or other diseases.

Seventh, we acknowledge that an average follow-up of about 4 years is relatively short for prostate cancer survival analysis. For this sample from the EMR, the actual follow-up time for each patient varies from 6 months to 10 years. Future studies can perform the analysis on larger multi-institutional datasets.

Eighth, we include a limited set of structured features as structured data such as diagnosis codes are often under-reported in the EMR. For example, we did not include the PSA scores because they are not well recorded in the structured data as many patients had PSA tests done at outside facilities[44]. We do include tumor grade in the structured data, which is shown to have a strong correlation to PSA[45]. We acknowledge this as a limitation of our study and future work can be done to augment the structured features.

In addition to future works, we also outline two limitations to applying our framework. First, our method can only uncover potential confounders that can be observed in notes. There are many sources of confounding in observational data and even rich EMR data cannot capture everything. If the confounding is unknown and unobservable, no method to our knowledge will be able to adjust for it. Hence, it would be good practice to perform sensitivity analysis to evaluate the result's robustness to unknown confounding. Please see Supplement 3 for additional discussion on the potential confounding situations we can capture.

Second, the validity of causal inference models cannot be determined without prospective experimental data. Therefore, the uncovered confounders and estimated HR can only be validated by clinicians. We are identifying potential candidates for the bias and then evaluating these candidates of bias against RCTs.

Many challenges remain for employing unstructured data for causal inference analysis and medical settings. We hope this work interests both clinical practitioners augmenting existing clinical support tools and researchers using textual data to reduce confounding in observational data. We hope our workflow, problem framing, and experimental design can serve as such a sandbox for testing more complex algorithms or adapting to other application areas. Ultimately, we hope this research will find causal information in clinical notes and provide a transparent way for machine learning to inform medical decision-making.

## Methods

**Dataset**. Our research conforms with all relevant ethical regulations and is approved by the Stanford Institutional Review Board (IRB). Patient consent was waived through obtaining the IRB. We curate a dataset of non-metastatic prostate and lung cancer patients from the Stanford Cancer Institute Research Database (SCIRDB). The database includes patients seen in the Stanford Health Care (SHC) system from 2008 to 2019 for prostate cancer and 2000 to 2019 for lung cancer. SHC clinical sites include one academic hospital, one freestanding cancer center, and several outpatient clinics. From SCIRDB, we pull a total of 3638 prostate cancer patients with 552,009 clinical notes and 3274 non-small cell lung cancer (NSCLC) patients with 648,505 clinical notes. The clinical notes include progress notes, letters, discharge summaries, emergency department notes, history and physical notes, and treatment planning notes.

For each patient, we also pull the structured EMR and data from the inpatient billing system. From the California Cancer Registry (CCR), we pull the available initial treatment information, cancer staging, tumor description, date of diagnosis, date of death, and date of the last follow-up for these selected patients. For NSCLC, we also pull the recorded Epic cancer staging information. Demographic information such as age, race, gender, and ethnicity are self-reported in our dataset.

**Study cohort**. We build our study cohorts from SCIRDB with reference to existing observational study principles and clinical expertise. We try our best to select patients for each treatment group built from the EMRs to match the RCTs criteria.

For each patient, we combine all treatments with the same Diagnosis ID in the CCR as the initial line of treatment. For patients with multiple Diagnosis ID, we

keep the first record of treatment. For prostate cancer, patients without a recorded treatment are labeled as active monitoring. To avoid explicit revelation of the treatment choice, we only include notes more than 2 months before the treatment start date for prostate cancer and 1 month for NSCLC. We rely on domain expertise to determine the 1 or 2-month pre-treatment cutoffs. Lung cancer patients typically have higher mortality and tend to start treatment pretty quickly. For prostate cancer, patients progress more slowly and get second opinions before making a treatment decision. We then select for patients with at least one note before the specified time. We select only patients who survived at least 6 months past their date of diagnosis to mitigate immortal-time bias[43]. We then filter for only patients with treatment of interest for analysis. Because we extract only initial treatments (rather than treatments for cancer recurrence) as recorded in SEER, most of the treatments are administered within 6 months of the diagnosis date[46]. This is similar to the setup for traditional landmark analysis[43]. To ensure the proportional hazard condition, patients who are still living are censored at the time of last follow-up[47]. The patient filtering and cohort selection process is shown in Fig. 4.

For patients with unknown clinical stage but known pathological stage, we impute the clinical stage by training a clinical-stage classification model using the pathological stage and other patient information. The pathological stage is usually a little higher than clinical stage due to the staging based on biopsy samples instead of imaging; hence, it is inaccurate to group them together. Clinical stage is more frequently used for similar observational studies[8,33] and it is more rigorous to impute the missing clinical stage with a model trained on the pathological stage and other relevant covariates. We train the clinical-stage imputation model with struct:patient_age, struct:pathological_stage, struct:diagnosis_year, and struct:tumor_grade. For NSCLC, text:tumor_grade is not included due to missing information. For both prostate and NSCLC, we train and validate a random forest model[48,49] on patients with both clinical and pathological stage available. The imputed stages are used as the clinical stage for those patients. For patients with both clinical and pathological stage missing, we are able to fill in some through clinical chart reviews.

We assign patients to the treatment groups based on the initial treatment decision to capture the intent to treat rather than the actual treatments administered. We assign patients with only surgery records into the surgery group and patients with only radiation records into the radiation group. For patients with both radiation and surgery, patients who received surgery first are assigned to the surgery group and patients who received radiation first to the radiation group. For prostate cancer, patients from all stages are included, except for patients with distant metastases, and patients with no recorded treatment are assigned to the active monitoring group. For NSCLC, only patients with clinical stage I are included. The data processing is performed in python with pandas[50].

**Data processing and representation**. We build the covariates used for uncovering confounders through the process shown in Fig. 1a. We compile the data from SCIRDB, CCR, and Epic for each patient.

We include age, race, ethnicity, clinical stage, and diagnosis year as part of the structured data. For prostate cancer, we also include SEER-recorded tumor grade, which are highly correlated with the Gleason grade. For NSCLC, we also include gender. Based on age range categories used in Li et al.[51], we form the categorical variable struct:patient_age by splitting age into ranges of ≤49 years old, 5-year buckets from 50–84 years old, and ≥85 years old. Race and ethnicity are encoded as one-hot vectors, with each feature indicating one race or ethnicity. Race is combined based on what is done in Li et al.[51]. We select these structured covariates because they are commonly accepted by clinicians as potential confounders and often included in CER studies[7]. For race, struct:race_unknown is not included as a

covariate. For ethnicity, only struct:hispanic is included as a covariate. For tumor grade, patients with unknown grade are imputed with the median grade value. The indicator variable struct:grade_unknown is added to indicate which patients have been imputed. The covariates struct:tumor_grade and struct:grade_unknown are not included for NSCLC due to missing information of tumor grade and clinical judgment. In the end, we have nine structured covariates for prostate cancer and seven structured covariates for NSCLC. While billing codes can be used to generate additional structured features for diagnosis and past treatments, existing studies have found these can be unreliable[27,28]. Hence, we chose to focus mainly on clinical notes to capture additional information that can influence survival time, such as patient symptoms and performance status.

We build word frequency representations of the clinical notes for the unstructured covariates. For each patient, we compile notes within the specified time (i.e., 2 months prior to the treatment start date for prostate cancer and 1 month prior for NSCLC). We only use notes from before treatment so that we are not predicting survival outcome with information unavailable at the time of treatment decision. The different time windows for the two diseases were selected as NSCLC treatment generally starts more quickly than prostate cancer treatment due to the more rapidly progressing nature of cancer. The notes are segmented based on clinical field labels (e.g., "IMPRESSION:", "HISTORY:"), tab spaces, NLTK sentence tokenization[52]. To remove noise, we remove clinical field labels and two sentences from the beginning and end of each document. We also remove sentences with common locations (e.g., "Stanford Medical Center", "Palo Alto") and medical doctor names (e.g., "xx xx, M.D.") as these are often prefix or suffix to note documents. To avoid including conditions patients do not have, we remove sentences if they contain less than 15 words including a negation term (i.e., "no", "denies", "does not", "none"). For example, this prevents us from extracting "smoking" as a covariate from "No history of smoking."

We then identify biomedical entities from the preprocessed clinical notes with scispaCy[53]. scispaCy is a spaCy[54]-based model for processing biomedical, scientific, and clinical text. The scispaCy models identifies a list of all the entities in the text that exist in a biomedical dictionary, such as the Unified Medical Language System[55]. We then lemmatize and combine all biomedical entities identified from the sentences for each patient into a single document. For lemmatization, we used the scispaCy lemmatizer, which is based on the spaCy lemmatization model. To further remove noise, we remove stopwords using a combination of the NLTK stopwords[52] and data-specific stopwords such as medical units (e.g., "lb", "oz", "mmhg"), time terms (e.g., "months", "days"), and medical or Stanford specific terms (e.g., "stanford", "patient", "doctor") that are very common but irrelevant to the task at hand. We also create a dictionary of synonyms in the dataset and use the dictionary to combine these words. The dictionary includes lexical variations that are not reduced to the same root during lemmatization (e.g., "abnormality" → "abnormal", "consult" → "consultation"), abbreviations (e.g., "hx" → "history", "fu" → "follow-up"), and common synonyms (e.g., "assistance → "service", "action" → "movement"). Please see a list of the synonyms included in the Appendix.

Finally, we remove punctuation and generate term frequency representations of the text using bag-of-words (BOW) with term frequency–inverse document frequency (TF-IDF) weighting[56]. Bag-of-words (BOW) model is a simplifying representation in natural language processing. It represents text (such as sentence or document) as a vector of word occurrence count. TF-IDF, is a score that reweighs the BOW matrix to reflect how important a word is to a document in a collection or corpus. We implement this with scikit-learn[49]. For prostate cancer, we select for the top 500 most frequent features using only unigrams. For NSCLC, we select for the top 1000 most frequent features using both unigrams and bigrams, and apply a document frequency threshold strictly lower than 0.7 to filter out dataset-specific stopwords. Although there are more prostate cancer patients, the lower number of death events makes it more difficult to include as many covariates

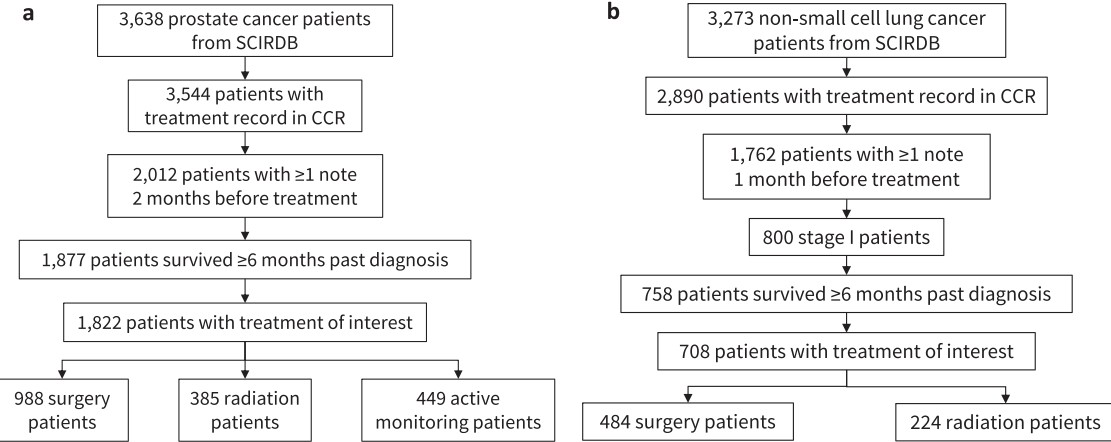

**Fig. 4 Patient cohort selection process for prostate and lung cancer patients. a** We show the cohort filtering process the the subsequent number of patients selected for prostate cancer. **b** We show the filtering process for NSCLC.

when performing survival analysis. Hence, we have 500 unstructured covariates for prostate cancer and 1000 unstructured covariates for NSCLC.

We scale and normalize both the structured and unstructured covariates before concatenating them. In total, we build 509 covariates for prostate cancer and 1007 covariates for NSCLC. These covariates are then used to uncover potential dataset-specific confounders.

**Outcomes**. We define our survival outcome as $(Y_i, E_i)$, where $Y_i \in \mathbb{Z}^+$ is the number of survival days since the diagnosis and $E_i \in \{0, 1\}$ is an indicator for whether a death event has been observed during follow-up. The treatment, $W_i \in \{0, 1\}$, is an indicator for either surgery, radiation, or monitoring, depending on the treatment group. The covariates, $X_i$, includes the structured dataset pulled from the EMR data and the bag-of-words matrix representation generated from EMR notes.

**Uncover and evaluate confounders**. We uncover interpretable potential confounders from the covariates and evaluate the confounders we've identified with survival analysis. The approach is shown in Fig. 1b.

We find the potential confounders by identifying covariates that are predictive of both treatment and survival outcome. We train prediction models for treatment ($W_i = 1$) and the survival outcome ($Y_i, E_i$) with Lasso[32] using glmnet[57]. Lasso is a $L1 -$ penalized linear regression that can produce coefficients for covariates that are exactly zero, and is, hence, often used for creating sparse models[58] or variable selection[15]. We select the intersection of covariates with non-zero coefficients from both the treatment and survival outcome models as potential confounders. For surgery vs. radiation and surgery vs. active monitoring for prostate cancer, we select the intersection covariates that correspond to the Lasso shrinkage penalty for the most regularized model such that the error is within one standard error of the minimum, *lambda.1se*. With radiation vs. monitoring for prostate cancer and surgery vs. radiation for stage I NSCLC, we select the intersection covariates that correspond to the shrinkage penalty that gives the minimum mean cross-validated error, *lambda.min*. The intersection terms selected are more stable with *lambda.1se*. However, we choose *lambda.min* for the latter two treatment groups because *lambda.1se* did not select any covariates from the text.

We then evaluate each of the covariate combinations with propensity score-adjusted survival analysis. Propensity scores for patient $i$ is the probability of receiving the treatment of interest, $W_i = 1$, given the covariates $X_i$[59]. Conditional on the propensity score, the distribution of observed covariates is expected to be the same in both branches of the treatment group. It is often used to reduce the effect of confounding in observational studies[59,60]. In survival analysis, the hazard rate $h(t|X)$ is the probability the patient will die within time $t$ given covariates $X$. The HR is the ratio of the hazard rate of the two treatments. In survival outcomes analysis, the HR is interpreted as the effect on survival for choosing the treatment of interest, $W_i = 1$.

We use the Cox-proportional hazard (Cox-PH) model to perform survival regression[61]. We assume the proportional hazards condition[62], which states that covariates are multiplicatively related to the hazard, e.g., a covariate may halve a subject's hazard at any given time $t$ while the baseline hazard may vary. Hence, the effect of covariates estimated by any proportional hazards model can be reported as the HR of the covariate.

In a Cox-PH model, the hazard rate of an individual is a linear function of their static covariates and a population-level baseline hazard that changes over time. We adjust for covariates (e.g., struct:patient_age, struct:race_white, etc.) against the duration of survival and a binary variable indicating whether the outcome event has occurred. We estimate

$$h(t|X) = h_0(t) \exp\left( b_w W + \sum_{j=1}^{p} b_j X_j \right), \quad (1)$$

where $p$ is the number of covariates, $h_0(t)$ the baseline hazard, $b_W$ the effect size of the treatment, and $b_j$ the effect size of the $j$th covariate. The HR for a covariate is equal to $e^{b_i}$. We define the HR of the treatment as $e^{b_w}$. The Lasso regularization can also be applied to a Cox-PH model for variable selection.

We use three methods to estimate the HR:

- Nearest-Neighbor Matching on Propensity Score (**matching**)[22]: We perform nearest-neighbor propensity score matching (NNM) on selected covariates and estimate the HR on the matched population using a univariate Cox-PH model regressed on the treatment.
- Inverse Propensity of Treatment Weighting (**IPTW**)[22,63]: We estimate the HR using a univariate Cox-PH model regressed on the treatment with inverse propensity score weighting with stabilization[63]. The weights are defined as

$$w_i = W_i + (1 - W_i)\left[ \frac{e(X_i)}{1 - e(X_i)} \right] \quad (2)$$

- Multivariate Cox proportional hazard (**multi.coxph**)[7,61,64]: We estimate the HR using a multivariate regression model on the treatment and selected covariates to see how covariates interact with each other. The multivariate model is also weighted with the inverse propensity scores above to form a doubly robust model.

All Cox-PH models are trained using the survival R package[65] with robust variance. Nearest-neighbor matching is performed using the MatchIt R package[66]. We estimate the propensity scores using logistic regression[67] with glmnet[57], stochastic gradient boosting[68] with gbm[69], and generalized random forests with grf[13]. We select the propensity score estimation method with the best overlap and covariate balance post propensity score adjustment.

We then compare the three methods for estimating HR using forest plots. For each covariate in struct+intersect, we also show the univariate and multivariate Cox-PH model HR, 95% HR confidence interval, and P value calculated using the Wald test from the survival R package[65]. Note that for the multivariate Cox-PH covariate analysis, we do not weight the model with the inverse propensity scores.

**Reporting summary**. Further information on research design is available in the Nature Research Reporting Summary linked to this article.

## Data availability
The datasets analyzed for the study are not publicly available. We extracted the data from the Stanford Cancer Institute Research Database, the California Cancer Registry, and the Epic System. The EHR data cannot be redistributed to researchers other than those approved through the Stanford Institutional Review Board and those who have obtained a Material Transfer Agreement. We have therefore given a detailed description of our data selection and processing pipeline in the Methods section. To request access to the data, please contact Jiaming Zeng at jiaming@alumni.stanford.edu.

## Code availability
Code for uncovering potential confounders after data processing is available at https://github.com/jmzeng/interpretable-potential-confounders[70]. A list of relevant packages used can be found in the README.

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

## Acknowledgements

For data acquisition, we also thank A. Solomon Henry and Douglas Wood. We also thank Allison Koenecke for providing a basis for the code and Ruoxuan Xiong for reviewing previous versions of the paper. The research is supported by funding from the Stanford Human-Centered Artificial Intelligence Institute and the Department of Management Science and Engineering. Athey thanks the Sloan Foundation and the Office of Naval Research grant ONR N00014-17-1-2131 for generous support.

## Author contributions

J.Z. contributed to data acquisition, data processing, study design, methodology, implementation, interpretation of results, and drafted and revised the paper. M.F.G. contributed to the acquisition of data, study design, interpretation of results, and revised the paper draft. D.L.R. contributed to the acquisition of data, provision of computational resources, methodology, and revised the paper draft. S.A. contributed to study design, methodology, interpretation of results, and revised the paper draft. R.D.S. contributed to the acquisition of data, study design, methodology, interpretation of results, and revised the paper draft. All authors contributed to the study conception, provided feedback during the work development, and gave approval for the submission.

## Competing interests

The authors declare no competing interests.
