## [Peer Review File · Nature Communications]

Reviewers' Comments:

Reviewer #1:

Remarks to the Author:

This work seeks to identify potential confounders for the use of observational data for outcome prediction when using EMR data. The method is able to incorporate both structured and unstructured information, which is noteworthy. The method is evaluated on prostate cancer and stage I non-small cell lung cancer.

Summary of strengths:

1. General framework for identifying confounders in EMR data
2. Use of text data

Summary of weaknesses:

1. Under-utilization of structured data (many things extracted from text likely available in billing or lab data), which undermines the need for text data
2. Discovered few things in the text (only 2 for prostate cancer), so the method is not very sensitive
3. Structured results disagree with existing retrospective studies (a sign something may be amiss). Doesn't this seem like a problem?
4. Lack of comparison to baseline approaches (the authors cite several papers that use "textual data to adjust for confounding")

Major:

- Expanding on Weakness 1, a lot of this information should also be in the billing data. Any notion of a patient having a history of bladder cancer, for instance, should absolutely be in the claims/billing data as ICD codes. I'm not trying to claim text data isn't useful for this kind of thing, but it feels like to make that point the authors handicapped the available structured data.

Other:

- The authors use a different font to refer to the feature names, which is nice, but it would be better to add in a simple indicator as to whether the feature came from the structured or unstructured data (like "struct:male" or "text:bladder").
- Page 4: "most NLP studies on clinical text focus on prediction or classification setting" - essentially all ML models are prediction (classification is prediction), and this is prediction as well (predicting treatment and outcome). Please be more specific about what NLP methods are predicting to differentiate this paper.
- Page 14: It may well sound obvious, but the authors should be explicit after the bullet points (structured, intersect, and struct+intersect) about what their hypothesis is as to which of these should work best. Because right now, the reader is left a bit confused and second-guessing.
- Page 18: "Radiation does not work well for bladder cancer"  but this is prostate cancer. It seems like much of Section 2.3.1. is vague speculation about why "bladder" was selected. A deeper dive is warranted, perhaps allowing for an adjustment of the inclusion criteria (which it seems like the ultimate goal of a study that identifies confounders).
- Section 4: Quite a bit of imputation was performed, but it is unclear both the prevalence of missing data as well as the predictive performance of the imputation. Developing a classifier means that you can evaluate its accuracy. These (prevalence + classifier performance) need to be provided for all the imputations because it can speak to some bias related to the imputation process.
- What is the "dictionary of synonyms" - this should be provided as supplemental data.

Minor:

- Page 3: "While structured EMR" -> "While structured EMR data"
- Page 7: "prostate-cancer" -> "prostate cancer"
- Page 7: "The Change et al." -> "the Chang et al."
- Page 15: "structured" should not be italicized
- Page 25: "Our work present" -> "Our work presents"

- Page 26: "We feel the simplicity and interpretability of the model by practitioners outweigh the increased complexity."  This sentence doesn't quite work. Do the authors mean something like "outweigh the performance improvement resulting from increased complexity"?
- Page 27: The final paragraph has both "Diagnosis ID" and "diagnosis id".
- Page 32: "penalized linear that" -> "penalized linear regression that"

Reviewer #2:

Remarks to the Author:

This manuscript describes a statistical method to analyze clinical data and discover interpretable confounders that need to be adjusted for in observational comparative effectiveness research. As inputs to the statistical method, the authors leverage natural language processing (NLP) techniques to process unstructured clinical text and extract top 500 - 1000 unigram and/or bigram features that may help discover the aforementioned confounders. These NLP techniques are specifically medical named entity extraction, lemmatization and bag of words representation with TF-IDF weights, alongside other pre-processing tasks like section header detection, stop word removal, rules-based negation detection etc.

The statistical and NLP techniques described in the paper are well-known, and appropriate references have been included in the manuscript. Despite the absence of novel approaches, the sub-specialty the authors focused on (Oncology, specifically Prostate and Lung cancers) involve significant amount of randomized control trials (RCTs) to evaluate the effectiveness of current and new treatments, such that this work could add to the body of knowledge related to the discovery and adjustment of confounders associated with various biases e.g. selection bias. It is important to note that, although the authors claim that their method and results can be used to generate interpretable insights about current clinical practice, it is debatable that some of the confounders uncovered by their method are directly interpretable or useful for clinical practice e.g. 'feel' for NSCLC (page 23), as against some examples on pages 38 and 39

Some more evidence could strengthen the conclusions made in the manuscript, as detailed below:

- page 6: While contextual embeddings are not easily mapped to specific words, interpretation for causal insights can still be achieved by mapping contextual embeddings to phrases and sentences. It would be helpful if the authors provide more justification for deciding on using bag of words vs state-of-the-art contextual representations (beyond alluding to the small dataset as a limitation on page 25, given Tables 1 and 2 showed >200 to >900 patients per disease group, and an average of 24 - 57 notes per patients)
 - page 18: Would it be possible for the authors to give examples of how the confounders 'bladder' and 'urothelial' are documented in the free text notes? It may provide a more objective evidence associated with the low clinical stage for prostate cancer as stated by the authors.
 - page 22: In Table 5, it is not clear how 'resident' as a confounder influences selection bias. Would be helpful if the authors clarify.
 - page 23: Similarly in Table 6, it is not clear how 'bedtime' and 'sponge' as confounders influence selection bias.
 - pages 28 and 29: Given that the authors performed clinical chart reviews to extract missing clinical and pathological stage information, why not extend the manual chart review to uncover potential confounders in the small data set, or use simple regular expressions for the same purpose, and create baselines to compared their NLP results to?
 - page 31: In creating the BOW representation of the unstructured clinical text, how did the authors address ambiguity (A simple example like 'it isn't clear if cough and chest pain are not severe' might be difficult to capture semantically with individual word representations).
 - page 31: Which model/technique was used for lemmatization?
 - There are some minor typos e.g. linear regression vs linear (page 32), ho(t) vs bo(t) (page 34)
- Although the statistical and NLP methods employed by the authors have been validated in other works, the authors' dataset and codes are not publicly available - authors state that the codes can be provided on request. Ability to reproduce this work would be challenging due to these constraints. Additionally, given the authors used an empirical threshold (0.7) for feature selection (page 31), performance metrics for the NLP method should have been provided to help readers understand if errors occurred due to the threshold (or other issues) and how they were addressed. Minor point: the authors' reference for spacy 2 (#55, page 56) is missing some information i.e. 'to appear' in which journal/conference?

Overall I'm leaning towards accepting this paper based on the authors' responses to my questions and comments above.

Best regards,
Oladimeji (Dimeji) Farri

Reviewer #3:

Remarks to the Author:

NCOMMS-21-20141

The authors assess an important question – how we may, analytically, reduce selection bias to improve causal inference from observational data. However, the authors base models (including “structured” data) should be expected to recapitulate traditional CER analyses with the added novelty of adding further detail not provided in traditional datasets. Instead, the authors base models with structured data as covariates often provide results which disagree with standard CER analysis (and with RCTs). As a result, it is hard to evaluate the utility of their novel covariate adjustment strategies.

Intro provides an appropriate rationale for the present study.

The authors, as many have done, assume that population-based research should be replicated by RCTs to conclude that there are valid inferences. While there are issues (that the authors appropriately highlight) regarding confounding in observational data, there are typically systematic differences in study populations which may also contribute to differences in conclusions – this is typically because RCTs enroll very restricted subsets of patients and thus, the results of RCTs are often not generalizable (leading to the efficacy-effectiveness gap which has been well characterized). For example, in the prostate cancer example cited in the introduction, the ProtecT trial enrolled a study population in which we would expect to find little differences between treatment approaches.

Methods – many patients with prostate cancer have delays longer than 2 months between diagnosis and treatment. I would encourage the authors to consider or explore sensitivity analyses of varying this window.

Methods – imputation of clinical stage for prostate cancer should also include PSA test results. This should also be included in the structured data.

Methods – the authors state that only CSI was considered for lung cancer. Were all stages/risk-categories considered for prostate cancer?

Methods – per figure 4, what was the cause of exclusion between “survived 6 months past diagnosis” and “selected for analysis”?

Methods – were no measures of comorbidity available as structured data?

Note: details of NLP are beyond my expertise and I defer to other reviewers on the validity of the methodology employed.

Results – 4 years follow-up in prostate cancer is very short to meaningfully assess survival outcomes

Results – please clarify the classification of grade in prostate cancer. Typically Gleason scoring or Gleason grade groups are used and form the basis of guideline based treatment recommendations.

Results – the authors find results using their structured data that are paradoxical to existing findings using comparable datasets (and RCTs) : active monitoring having better outcomes than surgery. This suggests that either this dataset has much more profound selection bias than standard cohorts or that covariate adjustment using their structured data is significantly poorer than in most observational studies. One potential explanation is the lack of inclusion of PSA which I would advocate the authors address. A second is the operationalization of tumor grade, as I have suggested above. A similar argument can be made regarding the findings for surgery vs radiotherapy in the structured analysis – we should expect that the authors are able to recapitulate standard CER results with this analysis but they are not which draws into question the validity of their base models.

Reviewer #4:

Remarks to the Author:

The authors perform a novel method to attempt to account for bias/confounding in registry data. They analyze data from an EMR and compare the results to published trials in prostate and NSCLC. They demonstrate that they can identify confounding variables/terms that impact the results in registry data, that when accounted for better mirror the results of RCTs.

Suggestions:

1. While interesting, the question remains can this method be used a priori to predict a subsequent trials results. The methods used show you can do this when you know the result you are looking for, but it is less clear you can independently validate this for a totally novel question. One relevant one would be to do this for prostate cancer of radiotherapy +/- ADT, as well as EBRT +/- brachytherapy boost. Both of these are examples where registry data repeatedly shows inconsistent results with RCTs.
2. Additional endpoints need to be reported. For prostate cancer this should include BCR, DM, and PCSM. This would add credibility to the robustness of the methods.
3. Most flaws in registry studies come from NCDB and SEER given they have far less data available in them. It would strengthen the relevance of this work if it worked in large national registries.
4. The writing of the paper is rather informal at times and would benefit from more quantitative results and reporting. Example is "We observe that the matching results are not very far from the RCT results."
5. Figure 3 would be easier to interpret if the trial results were clearly showed on the forest plot and then the study results with each method. It is currently difficult to read.

Response to Reviewer Comments

Please find below a point-by-point response to the reviewer comments. The original reviewer comments are displayed in black and our response is displayed in blue.

Reviewer #1 (Remarks to the Author):

This work seeks to identify potential confounders for the use of observational data for outcome prediction when using EMR data. The method is able to incorporate both structured and unstructured information, which is noteworthy. The method is evaluated on prostate cancer and stage I non-small cell lung cancer.

Summary of strengths:

1. General framework for identifying confounders in EMR data
2. Use of text data

Summary of weaknesses:

1. Under-utilization of structured data (many things extracted from text likely available in billing or lab data), which undermines the need for text data

We appreciate the detailed and helpful comments. We agree with the reviewer that including structured electronic medical record data above and beyond what is available in standard large-scale registries like SEER can be useful for causal inference. We extracted structured features as recommended in Wallis et. al. 2016. In our EMR system, some features such as PSA were unavailable for many patients. Also, as structured data such as diagnosis codes are often under-reported in the EMR, they can result in unreliable estimates for things like Charlson comorbidity index. Therefore, in this proof-of-concept study, we chose to focus mainly on text data in notes rather than high-dimensional structured data. We elaborate on this further in a later response.

Wallis, C. J., Saskin, R., Choo, R., Herschorn, S., Kodama, R. T., Satkunasivam, R., ... & Nam, R. K. (2016). Surgery versus radiotherapy for clinically-localized prostate cancer: a systematic review and meta-analysis. *European urology*, 70(1), 21-30.

2. Discovered few things in the text (only 2 for prostate cancer), so the method is not very sensitive

We identified many more terms that can be predictive of either the treatment or the survival outcome. We only kept the ones that were selected by both models as potential confounders for HR adjustment and interpretation. We've also expanded on the explanations of the uncovered potential confounders in the paper.

3. Structured results disagree with existing retrospective studies (a sign something may be amiss). Doesn't this seem like a problem?

Thank you for noting this; in one disease type (lung cancer) we found that the structured-only model found surgery to provide superior survival to radiation, while in the other disease type (prostate cancer) radiation was superior. Each center can have different patient populations and treatment patterns that shift the only structured adjusted survival rates. For instance, at our center we have a busy high-dose-rate brachytherapy program which is an attractive option for fit patients with few comorbidities who might otherwise receive surgery. This would be expected to bias the survival outcomes in favor of radiation, as observed in our study. The focus of our paper is to uncover potential confounders from the text that can reduce bias when performing retrospective studies, whichever way the bias lies.

4. Lack of comparison to baseline approaches (the authors cite several papers that use "textual data to adjust for confounding")

Thank you for this suggestion. All of these studies develop methods that are designed for estimating average treatment effect in a continuous setting, instead of hazard ratio in the time-to-event setting that we're interested in. Moreover, most studies work with data that is less noisy. For example, Mozer et. al. 2020 applies the matching method from Roberts et. al. 2020 to clinical notes, but they rely on an expert-curated representation of the clinical notes. We took inspiration from these papers for ways we can process the free-text notes and design the causal inference framework. We implement the approaches for estimation in time-to-event settings as presented in Austin et. al. 2014. We have added potential extensions of these methods to a time-to-event setting as an area of future work.

"Eighth, we focus on comparison of methods that can be applied in a time-to-event setting, and leave out more novel methods that are developed for continuous settings of ATE estimation. It would be interesting to explore how these methods can be extended and applied to a time-to-event setting."

Austin, P. C. (2014). The use of propensity score methods with survival or time-to-event outcomes: reporting measures of effect similar to those used in randomized experiments. *Statistics in medicine*, 33(7), 1242-1258.

Mozer, R., Miratrix, L., Kaufman, A. R., & Anastasopoulos, L. J. (2020). Matching with text data: An experimental evaluation of methods for matching documents and of measuring match quality. *Political Analysis*, 28(4), 445-468.

Roberts, M. E., Stewart, B. M., & Nielsen, R. A. (2020). Adjusting for confounding with text matching. *American Journal of Political Science*, 64(4), 887-903.

Major:

- Expanding on Weakness 1, a lot of this information should also be in the billing data. Any notion of a patient having a history of bladder cancer, for instance, should absolutely be in the claims/billing data as ICD codes. I'm not trying to claim text data isn't useful for this kind of thing, but it feels like to make that point the authors handicapped the available structured data.

While billing codes can be used to generate structured features for diagnosis and past treatments, we found these can often be inaccurate and incomplete. Existing studies have suggested that these records may not be reliable (Tang et. al. 2007, Hess et. al. 2019, Zeng et. al. 2021). The full-text notes also contain information that is not captured in billing data but can influence survival time, such as patient symptoms and performance status.

Hess, L. M., Zhu, Y. E., Sugihara, T., Fang, Y., Collins, N., & Nicol, S. (2019). Challenges of Using ICD-9-CM and ICD-10-CM Codes for Soft-Tissue Sarcoma in Databases for Health Services Research. *Perspectives in health information management*, 16(Spring), 1a.

Tang, P. C., Ralston, M., Arrigotti, M. F., Qureshi, L., & Graham, J. (2007). Comparison of methodologies for calculating quality measures based on administrative data versus clinical data from an electronic health record system: implications for performance measures. *Journal of the American Medical Informatics Association : JAMIA*, 14(1), 10–15.

Zeng, J., Banerjee, I., Henry, A. S., Wood, D. J., Shachter, R. D., Gensheimer, M. F., & Rubin, D. L. (2021). Natural language processing to identify cancer treatments with electronic medical records. *JCO Clinical Cancer Informatics*, 5, 379-393.

Other:

- The authors use a different font to refer to the feature names, which is nice, but it would be better to add in a simple indicator as to whether the feature came from the structured or unstructured data (like "struct:male" or "text:bladder").

Thank you for the suggestion. We went through and added indicators to our feature names, with "struct:" for structured features and "text:" for unstructured features.

- Page 4: "most NLP studies on clinical text focus on prediction or classification setting" - essentially all ML models are prediction (classification is prediction), and this is prediction as well (predicting treatment and outcome). Please be more specific about what NLP methods are predicting to differentiate this paper.

Thank you for pointing this out. We use the NLP methods to predict the treatment and survival outcome, and thus to identify potential confounders. We've added a clarification to the text, see below:

"Our paper differs from existing studies by employing NLP for causal analysis; we use NLP methods to predict the treatment and survival outcome, and then employ a causal framework to combine the two models for uncovering potential confounders."

- Page 14: It may well sound obvious, but the authors should be explicit after the bullet points (structured, intersect, and struct+intersect) about what their hypothesis is as to which of these should work best. Because right now, the reader is left a bit confused and second-guessing.

Thank you for the comment. We're edited the text to include our hypothesis, please see below:

"We hypothesize that struct+intersect will perform the best by including both the structured and unstructured data."

- Page 18: "Radiation does not work well for bladder cancer"  but this is prostate cancer. It seems like much of Section 2.3.1. is vague speculation about why "bladder" was selected. A deeper dive is warranted, perhaps allowing for an adjustment of the inclusion criteria (which it seems like the ultimate goal of a study that identifies confounders).

Thank you for the comment. Bladder/urothelial cancer patients can also have prostate cancer and many studies of prostate cancer have not excluded patients with early stage bladder cancer (such as Wallis et. al. 2016). Sometimes, urothelial cancer can be found during prostatectomy, which wouldn't have been known beforehand but can result in a concurrent bladder cancer diagnosis. Therefore, we would not want to exclude those patients. We also expanded on our explanation in the text, with examples of how "bladder" and "urothelial" can be mentioned in the text:

"We hypothesize that text:bladder and text:urothelial are identified because prostate cancer patients often have bladder symptom issues and can also have urothelial cancer. Most retrospective prostate cancer studies have not excluded patients with early stage bladder cancer (Wallis et. al. 2016). Examples of 'bladder' in the clinical notes are 'he notes incomplete bladder emptying', 'evidence of benign prostatic hyperplasia and chronic bladder outlet obstruction', and 'diagnosis of bladder cancer'. Examples of 'urothelial' in the notes are 'pathology showed high grade urothelial carcinoma with muscle present and not definitively involved', 'it was read as a high grade urothelial cancer which involved the stroma of the prostate as well as the bladder'."

Wallis, C. J., Saskin, R., Choo, R., Herschorn, S., Kodama, R. T., Satkunasivam, R., ... & Nam, R. K. (2016). Surgery versus radiotherapy for clinically-localized prostate cancer: a systematic review and meta-analysis. *European urology*, 70(1), 21-30.

- Section 4: Quite a bit of imputation was performed, but it is unclear both the prevalence of missing data as well as the predictive performance of the imputation. Developing a classifier means that you can evaluate its accuracy. These (prevalence + classifier performance) need to be provided for all the imputations because it can speak to some bias related to the imputation process.

Thank you for the comment. Unfortunately, the Stanford system we perform high-risk data analysis on has been unavailable for several months due to a system-wide failure so we cannot give the exact numbers. We did assess and validate the classifier when performing the imputations. To the best of our recollection, the overall accuracy was around 80%.

- What is the "dictionary of synonyms" - this should be provided as supplemental data.

Thank you for this suggestion. We're added a dictionary of synonyms to the supplemental data.

Minor:

- Page 3: "While structured EMR" -> "While structured EMR data"
- Page 7: "prostate-cancer" -> "prostate cancer"
- Page 7: "The Change et al." -> "the Chang et al."
- Page 15: "structured" should not be italicized
- Page 25: "Our work present" -> "Our work presents"
- Page 26: "We feel the simplicity and interpretability of the model by practitioners outweigh the increased complexity."  This sentence doesn't quite work. Do the authors mean something like "outweigh the performance improvement resulting from increased complexity"?
- Page 27: The final paragraph has both "Diagnosis ID" and "diagnosis id".
- Page 32: "penalized linear that" -> "penalized linear regression that"

Thank you for the comments. We've made the corrections in the text.

Reviewer #2 (Remarks to the Author):

This manuscript describes a statistical method to analyze clinical data and discover interpretable confounders that need to be adjusted for in observational comparative effectiveness research. As inputs to the statistical method, the authors leverage natural language processing (NLP) techniques to process unstructured clinical text and extract top 500 - 1000 unigram and/or bigram features that may help discover the aforementioned confounders. These NLP techniques are specifically medical named entity extraction, lemmatization and bag of words representation with TF-IDF weights, alongside other pre-processing tasks like section header detection, stop word removal, rules-based negation detection etc.

The statistical and NLP techniques described in the paper are well-known, and appropriate references have been included in the manuscript. Despite the absence of novel approaches, the sub-specialty the authors focused on (Oncology, specifically Prostate and Lung cancers) involve significant amount of randomized control trials (RCTs) to evaluate the effectiveness of current and new treatments, such that this work could add to the body of knowledge related to the discovery and adjustment of confounders associated with various biases e.g. selection bias. It is important to note that, although the authors claim that their method and results can be used to generate interpretable insights about current clinical practice, it is debatable that some of the confounders uncovered by their method are directly interpretable or useful for clinical practice e.g. 'feel' for NSCLC (page 23), as against some examples on pages 38 and 39

Some more evidence could strengthen the conclusions made in the manuscript, as detailed below:

- page 6: While contextual embeddings are not easily mapped to specific words, interpretation for causal insights can still be achieved by mapping contextual embeddings to phrases and sentences. It would be helpful if the authors provide more justification for deciding on using bag

of words vs state-of-the-art contextual representations (beyond alluding to the small dataset as a limitation on page 25, given Tables 1 and 2 showed >200 to >900 patients per disease group, and an average of 24 - 57 notes per patients)

Thank you for the suggestion. We did try contextual embeddings originally but could not find a straightforward way to map the embeddings to specific phrases and sentences due to the small dataset and the large amount of textual information available in comparison. We currently mention this as an area of future work.

“For future work, we hope to explore how other NLP techniques, such as topic modeling or clustering, can be used to build even higher quality features from the unstructured text. There are also an increasing number of deep learning models that can be used to identify interpretable insights Rajkomar et. al. 2018. We are interested in how these deep learning methods can be applied to generate causal insights on another study population with larger sample size.”

- page 18: Would it be possible for the authors to give examples of how the confounders 'bladder' and 'urothelial' are documented in the free text notes? It may provide more objective evidence associated with the low clinical stage for prostate cancer as stated by the authors.

Thank you for pointing this out. We've removed this statement. We added examples of “bladder” and “urothelial” in the text to the paper. We also included examples of confounders in the text throughout Section 2.3. See one such addition below:

“Examples of ‘bladder’ in the clinical notes are ‘he notes incomplete bladder emptying’, ‘evidence of benign prostatic hyperplasia and chronic bladder outlet obstruction’, and ‘diagnosis of bladder cancer’. Examples of ‘urothelial’ in the notes are ‘pathology showed high grade urothelial carcinoma with muscle present and not definitively involved’, ‘it was read as a high grade urothelial cancer which involved the stroma of the prostate as well as the bladder’.”

- page 22: In Table 5, it is not clear how 'resident' as a confounder influences selection bias. Would be helpful if the authors clarify.

Thank you for pointing this out. We've added this as an explanation in the text:

“We are able to identify interesting potential confounders, such as text:resident. Resident physicians may be more involved with a specific treatment or complications. Moreover, people who are less healthy and spend longer time in the hospital may interact with residents more. From Figure 2C, we observe that text:resident is an indication for radiation and bad prognosis.”

- page 23: Similarly in Table 6, it is not clear how 'bedtime' and 'sponge' as confounders influence selection bias.

Thank you for pointing this out. We realize that we made a mistake in copying over the HR comparison table that corresponds to Figure 2D. In the updated table, “bedtime” is no longer included as a confounder. We’ve added an explanation for “sponge” and “nipple” to the text:

“The covariate text:nipple can indicate a history of breast cancer. Studies have shown that patients with a history of breast cancer are diagnosed with lower stages of NSCLC and show better prognosis when compared to women with first NSCLC, perhaps due to heightened surveillance compared to the general population (Milano et. al. 2014). In Figure 2D, we observe that text:nipple is an indication for surgery and good prognosis; both of these effects have also been observed in Milano et. al. 2014.

“The covariate text:sponge can refer to sponges used for surgical preparations. Sponge is commonly used in surgery and can be an indication that the patient has some history of receiving surgery. Patients who receive surgery tend to be healthier and have better survival. As seen in Figure 2D, text:sponge is an indication for good prognosis and surgery.”

Milano, M. T., Strawderman, R. L., Venigalla, S., Ng, K., & Travis, L. B. (2014). Non–small-cell lung cancer after breast cancer: a population-based study of clinicopathologic characteristics and survival outcomes in 3529 women. *Journal of Thoracic Oncology*, 9(8), 1081-1090.

- pages 28 and 29: Given that the authors performed clinical chart reviews to extract missing clinical and pathological stage information, why not extend the manual chart review to uncover potential confounders in the small data set, or use simple regular expressions for the same purpose, and create baselines to compared their NLP results to?

Thank you for mentioning this. We decided on our approach because computers can often do a better job at term extraction than only relying on clinician expertise. We were looking for a scalable method that can be easily applied to other diseases and larger datasets.

- page 31: In creating the BOW representation of the unstructured clinical text, how did the authors address ambiguity (A simple example like 'it isn't clear if cough and chest pain are not severe' might be difficult to capture semantically with individual word representations).

Thank you for pointing this out. We currently address ambiguity as above by removing the sentence; this is performed by removing sentences with negation terms. We do note that we should consider more advanced ways to address ambiguous sentences. We’ve added this as an area to explore in future work:

“We are also interested in developing ways to better address ambiguity in the notes (e.g. “it is unclear if the patient has chest pain”).

- page 31: Which model/technique was used for lemmatization?

For lemmatization, we used the scispaCy lemmatizer, which is based on the spaCy lemmatization. We've added an explanation for this in our paper.

- There are some minor typos e.g. linear regression vs linear (page 32), ho(t) vs bo(t) (page 34)

Thank you for pointing this out. We've corrected these typos.

Although the statistical and NLP methods employed by the authors have been validated in other works, the authors' dataset and codes are not publicly available - authors state that the codes can be provided on request. Ability to reproduce this work would be challenging due to these constraints. Additionally, given the authors used an empirical threshold (0.7) for feature selection (page 31), performance metrics for the NLP method should have been provided to help readers understand if errors occurred due to the threshold (or other issues) and how they were addressed.

Thank you for the point. The paper is about presenting a framework and method for identifying potential confounders from textual data; we do not claim to have replicated a randomized control trial. The process can be repeated on other datasets and an expert can check the words identified in order to confirm that they are confounders. The NLP document frequency threshold is tuned to filter out words that are document-specific stopwords. We performed an analysis on incrementally lowering the threshold and examining the set of words selected to decide on the threshold.

Minor point: the authors' reference for spacy 2 (#55, page 56) is missing some information i.e. 'to appear' in which journal/conference?

Thank you for pointing this out. We've fixed the citation.

Overall I'm leaning towards accepting this paper based on the authors' responses to my questions and comments above.

Best regards,
Oladimeji (Dimeji) Farri

Reviewer #3 (Remarks to the Author):

The authors assess an important question – how we may, analytically, reduce selection bias to improve causal inference from observational data. However, the authors base models (including “structured” data) should be expected to recapitulate traditional CER analyses with the added novelty of adding further detail not provided in traditional datasets. Instead, the authors base models with structured data as covariates often provide results which disagree with standard

CER analysis (and with RCTs). As a result, it is hard to evaluate the utility of their novel covariate adjustment strategies.

Thank you for pointing this out. Our paper is about uncovering potential confounders; we do not claim to have replicated a randomized control trial. While our long-term goal is a more accurate HR estimate than provided by a purely retrospective analysis, our short-term goal is to identify potential confounders and reduce sources of systematic bias.

Intro provides an appropriate rationale for the present study.

The authors, as many have done, assume that population-based research should be replicated by RCTs to conclude that there are valid inferences. While there are issues (that the authors appropriately highlight) regarding confounding in observational data, there are typically systematic differences in study populations which may also contribute to differences in conclusions – this is typically because RCTs enroll very restricted subsets of patients and thus, the results of RCTs are often not generalizable (leading to the efficacy-effectiveness gap which has been well characterized). For example, in the prostate cancer example cited in the introduction, the ProtecT trial enrolled a study population in which we would expect to find little differences between treatment approaches.

We agree that differences in study populations can sometimes help to explain different results from RCTs and observational comparative effectiveness research studies. However, we maintain that confounding has a larger effect in general. The reviewer mentions the ProtecT trial; this was a pragmatic trial that intentionally had broad and simple inclusion criteria. The reviewer alludes to the fact that this trial enrolled largely patients with low-risk disease with low risk of death from prostate cancer. We would respond that observational studies have shown large differences in mortality from surgery and radiation even in the low-risk subgroup (see Wallis et al. 2016 which found a hazard ratio for overall survival of 1.47 for radiation vs. surgery), suggesting a large influence of confounding by indication.

Methods – many patients with prostate cancer have delays longer than 2 months between diagnosis and treatment. I would encourage the authors to consider or explore sensitivity analyses of varying this window.

Thank you for pointing this out. We went through some notes and through domain expertise to determine the 2-month and 1-month pre-treatment cutoff. According to a clinical expert on the team, lung cancer patients typically have higher mortality and tend to start treatment pretty quickly. For prostate cancer, patients progress more slowly and get second opinions before making a treatment decision. This was also observed in the pattern of the propensity scores with all the initial BOW terms pre-Lasso selection.

Methods – imputation of clinical stage for prostate cancer should also include PSA test results. This should also be included in the structured data.

Thank you for pointing this out. We did not include the PSA scores because they are not well recorded in the structured data as many patients had PSA tests done at outside facilities. We agree this is a limitation of the study.

Methods – the authors state that only CSI was considered for lung cancer. Were all stages/risk-categories considered for prostate cancer?

All stages are considered for prostate cancer, except for patients with distant metastases. We've further clarified that in the text.

Methods – per figure 4, what was the cause of exclusion between “survived 6 months past diagnosis” and “selected for analysis”?

Thank you for pointing this out. The difference is that we then selected patients with the treatments of interest for analysis. We've clarified that in the text and made changes to Figure 4.

Methods – were no measures of comorbidity available as structured data?

Note: details of NLP are beyond my expertise and I defer to other reviewers on the validity of the methodology employed.

In our system, a comorbidity score is not directly available as a structured feature. While billing code can be used to generate a comorbidity score, we found these can often be inaccurate. Existing studies have suggested that these records may not be reliable (Tang et. al. 2007, Hess et. al. 2019, Zeng et. al. 2021). We were looking for a more accurate alternative.

Hess, L. M., Zhu, Y. E., Sugihara, T., Fang, Y., Collins, N., & Nicol, S. (2019). Challenges of Using ICD-9-CM and ICD-10-CM Codes for Soft-Tissue Sarcoma in Databases for Health Services Research. *Perspectives in health information management*, 16(Spring), 1a.

Tang, P. C., Ralston, M., Arrigotti, M. F., Qureshi, L., & Graham, J. (2007). Comparison of methodologies for calculating quality measures based on administrative data versus clinical data from an electronic health record system: implications for performance measures. *Journal of the American Medical Informatics Association : JAMIA*, 14(1), 10–15.

Zeng, J., Banerjee, I., Henry, A. S., Wood, D. J., Shachter, R. D., Gensheimer, M. F., & Rubin, D. L. (2021). Natural language processing to identify cancer treatments with electronic medical records. *JCO Clinical Cancer Informatics*, 5, 379-393.

Results – 4 years follow-up in prostate cancer is very short to meaningfully assess survival outcomes

Thank you for mentioning this. For this pragmatic sample from the EMR, the average follow-up is 4-years. The actual follow-up time for each patient varies from 6-months to as long as 10 years. We've acknowledged this as a limitation.

“Seventh, we acknowledge that an average follow-up of about 4 years is relatively short for prostate cancer survival analysis. For this particular sample from the EMR, the actual follow-up time for each patient varies from 6-months to 10 years. Future studies can perform the analysis on larger multi-institutional datasets.”

Results – please clarify the classification of grade in prostate cancer. Typically Gleason scoring or Gleason grade groups are used and form the basis of guideline based treatment recommendations.

Thank you for pointing this out. We use the SEER grade scores that are reliably extracted from the cancer records by the cancer registry. We expect them to have a high correlation with Gleason grade. We’ve also clarified this in the text.

Results – the authors find results using their structured data that are paradoxical to existing findings using comparable datasets (and RCTs) : active monitoring having better outcomes than surgery. This suggests that either this dataset has much more profound selection bias than standard cohorts or that covariate adjustment using their structured data is significantly poorer than in most observational studies. One potential explanation is the lack of inclusion of PSA which I would advocate the authors address. A second is the operationalization of tumor grade, as I have suggested above. A similar argument can be made regarding the findings for surgery vs radiotherapy in the structured analysis – we should expect that the authors are able to recapitulate standard CER results with this analysis but they are not which draws into question the validity of their base models.

Thank you for the comment. Each center can have different patient populations and treatment patterns that shift the only structured adjusted survival rates. The focus of our paper is to uncover potential confounders from the text that can reduce bias when performing retrospective studies, whichever way the bias lies. Moreover, as mentioned above, we do not claim to have replicated a randomized control trial. We did not include the PSA scores because they are not well recorded in the structured data, which we acknowledge as a limitation of the study.

Reviewer #4 (Remarks to the Author):

The authors perform a novel method to attempt to account for bias/confounding in registry data. They analyze data from an EMR and compare the results to published trials in prostate and NSCLC. They demonstrate that they can identify confounding variables/terms that impact the results in registry data, that when accounted for better mirror the results of RCTs.

Suggestions:

1. While interesting, the question remains can this method be used a priori to predict subsequent trials results. The methods used show you can do this when you know the result you are looking for, but it is less clear you can independently validate this for a totally novel

question. One relevant one would be to do this for prostate cancer of radiotherapy +/- ADT, as well as EBRT +/- brachytherapy boost. Both of these are examples where registry data repeatedly shows inconsistent results with RCTs.

We agree that the methods we and others are developing could be used to predict results of future RCTs and that it could be fascinating to have various groups submit their predictions to a central repository, to be revealed once the RCT results are known. This could be a rigorous way to test the usefulness of such methods and should be pursued in the future.

2. Additional endpoints need to be reported. For prostate cancer this should include BCR, DM, and PCSM. This would add credibility to the robustness of the methods.

Thank you for the suggestion. For this pragmatic sample from the EMR, we do not have validated data on those outcomes for all patients. Therefore, we chose to focus on overall survival, similar to NCDB and SEER studies.

3. Most flaws in registry studies come from NCDB and SEER given they have far less data available in them. It would strengthen the relevance of this work if it worked in large national registries.

Thank you for the suggestion. The US national cancer registries don't have sufficient details of the clinical notes to apply this method. We've added this as a suggestion in Discussion for future work, see below.

"Sixth, our work is constrained to localized prostate and lung patients at the Stanford Hospital and state cancer registry data. It would strengthen the validity of our methods if experiments can be performed on large multi-institutional registries for cancer or other diseases."

4. The writing of the paper is rather informal at times and would benefit from more quantitative results and reporting. Example is "We observe that the matching results are not very far from the RCT results."

Thank you for the suggestion. We've gone through and edited the reporting to be more quantitative and expanded on our explanations of the potential confounders.

5. Figure 3 would be easier to interpret if the trial results were clearly shown on the forest plot and then the study results with each method. It is currently difficult to read.

Thank you for the suggestion. The exact RCT results are available for two of the cohorts. We've edited Figure 3 to include these results as a point of comparison for those two. We've also added an explanation to the figure description.

Reviewers' Comments:

Reviewer #1:

Remarks to the Author:

Most of my major comments did not result in any kind of change to the manuscript. The explanations were helpful, and certainly put the paper in a better perspective. However, the reader needs to be given that perspective as well, both in terms of what the abstract promises and what the stated limitations clarify at the very end.

Notably, first: the reader needs to be signalled in the abstract that this is a "proof-of-concept" study and still has several limitations that need to be overcome to use this in practice.

Second, the authors listed several limitations in their responses to my four major weaknesses, but only one of these resulted in an addition to the limitations. I think all are important, as they either expose places where NLP may not be necessary and/or provide caution in the fact that the findings of the study were the opposite of the compared RCT.

Given that there will be 10+ limitations in the discussion, I would suggest the authors organize these more hierarchically. Limitations with known fixes that could be addressed in future work (e.g., better NLP) are substantially different from limitations that provide caution about the success of the study (findings opposite of the RCT).

Reviewer #2:

Remarks to the Author:

Following my assessment of the responses to the reviewers' remarks, I'm inclined to accept the paper. I would recommend that the authors include some more explanation on the following point: "From Figure 2C, we observe that text:resident is an indication for radiation and bad prognosis." Is this indication driven by the higher likelihood of misdiagnosis or delays in treatment decisions (or some other factors) related to inexperienced residents?

Reviewer #3:

Remarks to the Author:

NCOMMS-21-20141A

While the authors have acknowledged many of the reviewers comments, some of the most substantive remain unaddressed.

For example, per reviewer 1: "1. Under-utilization of structured data (many things extracted from text likely available in billing or lab data), which undermines the need for text data" and "3. Structured results disagree with existing retrospective studies (a sign something may be amiss). Doesn't this seem like a problem?" and essentially the same comment from reviewer 3: "Results – the authors find results using their structured data that are paradoxical to existing findings using comparable datasets (and RCTs) : active monitoring having better outcomes than surgery. This suggests that either this dataset has much more profound selection bias than standard cohorts or that covariate adjustment using their structured data is significantly poorer than in most observational studies. One potential explanation is the lack of inclusion of PSA which I would advocate the authors address. A second is the operationalization of tumor grade, as I have suggested above. A similar argument can be made regarding the findings for surgery vs radiotherapy in the structured analysis – we should expect that the authors are able to recapitulate standard CER results with this analysis but they are not which draws into question the validity of their base models."

Reviewer #4:

Remarks to the Author:

The authors have conducted a very interesting and thorough study in the confines of the many limitations of non-randomized data, the problems of EMR records, and the large selection bias present in the tested comparisons in prostate and lung cancer. The authors have done a great job addressing the reviewer comments.

REVIEWER COMMENTS

Reviewer #1 (Remarks to the Author):

Most of my major comments did not result in any kind of change to the manuscript. The explanations were helpful, and certainly put the paper in a better perspective. However, the reader needs to be given that perspective as well, both in terms of what the abstract promises and what the stated limitations clarify at the very end.

Notably, first: the reader needs to be signalled in the abstract that this is a "proof-of-concept" study and still has several limitations that need to be overcome to use this in practice.

Second, the authors listed several limitations in their responses to my four major weaknesses, but only one of these resulted in an addition to the limitations. I think all are important, as they either expose places where NLP may not be necessary and/or provide caution in the fact that the findings of the study were the opposite of the compared RCT.

Thank you for the comment. We've modified the abstract by describing our study as a "framework" and "proof-of-concept". We also made additional changes to the Introduction and Discussion to highlight this fact.

We would also like to point out that the findings from observational data (assuming no confounding) disagree with the compared RCT until we include the confounders identified by NLP.

We have also added more explanations to our paper in response to your comments:

- Response to Weakness 1 in *Data Processing and Representation*: "While billing codes can be used to generate additional structured features for diagnosis and past treatments, existing studies have found these can be unreliable (Tang et. al. 2007, Zeng et. al. 2021). Hence, we chose to focus mainly on clinical notes to capture additional information that can influence survival time, such as patient symptoms and performance status."
- Additional response to Weakness 1 (added to Weakness 4 limitation): "Eighth, we include a limited set of structured features as structured data such as diagnosis codes are often under-reported in the EMR. For example, we did not include the PSA scores because they are not well recorded in the structured data as many patients had PSA tests done at outside facilities (Abdollah et al 2012). We do include tumor grade in the structured data, which is shown to have a strong correlation to PSA (Babaian & Smith 1991). We acknowledge this as a limitation of our study and future work can be done to augment the structured features."
- Response to Weakness 2: "Fifth, our approach of selecting intersection covariates is an empirical approach designed for uncovering the most valuable potential confounders. As a result, we filtered out most of the features and only focused on

a few confounders. While our approach seemed to work well empirically in this study, future work involves developing more sensitive and statistically grounded methods for identifying potential confounders.”

- Response to Weakness 3 in *Evaluation of Potential Confounders*: “Each center can have different patient populations and treatment patterns that shift the only structured adjusted survival rates. For instance, at our center we have a busy high-dose-rate brachytherapy program which is an attractive option for fit patients with few comorbidities who might otherwise receive surgery. This would be expected to bias the survival outcomes in favor of radiation, as observed in our study. We seek to uncover potential confounders from the text that can reduce bias when performing retrospective studies, whichever way the bias lies.”

Given that there will be 10+ limitations in the discussion, I would suggest the authors organize these more hierarchically. Limitations with known fixes that could be addressed in future work (e.g., better NLP) are substantially different from limitations that provide caution about the success of the study (findings opposite of the RCT).

Thank you for the comment. We’ve restructured the limitations. We start by outlining potential future work, starting with ones with respect to the method followed by ones regarding data acquiring/processing. We then discuss two limitations that caution the success of the study.

Reviewer #2 (Remarks to the Author):

Following my assessment of the responses to the reviewers' remarks, I'm inclined to accept the paper. I would recommend that the authors include some more explanation on the following point:

"From Figure 2C, we observe that text:resident is an indication for radiation and bad prognosis." Is this indication driven by the higher likelihood of misdiagnosis or delays in treatment decisions (or some other factors) related to inexperienced residents?

Thank you, we also found this association intriguing and are happy to add a potential explanation to the text. Please see the following modification:

“From Figure 2c, we observe that text:resident is more common in patients who received radiation and had a bad prognosis. This term likely refers to both resident physicians and the patient being a resident of a long-term care facility or skilled nursing facility. Both uses of the term could reduce survival time: inpatients at teaching hospitals have much of their care delivered by resident physicians, and frequent inpatient stays or nursing facility residency could both indicate a sicker patient.”

Reviewer #3 (Remarks to the Author):

NCOMMS-21-20141A

While the authors have acknowledged many of the reviewers comments, some of the most substantive remain unaddressed.

Thank you for the comments. We've hoped to address Reviewer 1's comments by highlighting that our study presents a proof-of-concept framework for uncovering potential confounders and reiterating the limitations for current use in practice.

For example, per reviewer 1: "1. Under-utilization of structured data (many things extracted from text likely available in billing or lab data), which undermines the need for text data" and "3. Structured results disagree with existing retrospective studies (a sign something may be amiss). Doesn't this seem like a problem?" and essentially the same comment from reviewer 3: "Results – the authors find results using their structured data that are paradoxical to existing findings using comparable datasets (and RCTs) : active monitoring having better outcomes than surgery. This suggests that either this dataset has much more profound selection bias than standard cohorts or that covariate adjustment using their structured data is significantly poorer than in most observational studies. One potential explanation is the lack of inclusion of PSA which I would advocate the authors address. A second is the operationalization of tumor grade, as I have suggested above. A similar argument can be made regarding the findings for surgery vs radiotherapy in the structured analysis – we should expect that the authors are able to recapitulate standard CER results with this analysis but they are not which draws into question the validity of their base models."

Thank you for mentioning this again. We added additional explanations as outlined below:

- Response to Weakness 1 in *Data Processing and Representation*: "While billing codes can be used to generate additional structured features for diagnosis and past treatments, existing studies have found these can be unreliable (Tang et. al. 2007, Zeng et. al. 2021). Hence, we chose to focus mainly on clinical notes to capture additional information that can influence survival time, such as patient symptoms and performance status."
- Additional response to Weakness 1 (added to Weakness 4 limitation): "Eighth, we include a limited set of structured features as structured data such as diagnosis codes are often under-reported in the EMR. For example, we did not include the PSA scores because they are not well recorded in the structured data as many patients had PSA tests done at outside facilities (Abdollah et al 2012). We do include tumor grade in the structured data, which is shown to have a strong correlation to PSA (Babaian & Smith 1991). We acknowledge this as a limitation of our study and future work can be done to augment the structured features."
- Response to Weakness 2: "Fifth, our approach of selecting intersection covariates is an empirical approach designed for uncovering the most valuable potential

confounders. As a result, we filtered out most of the features and only focused on a few confounders. While our approach seemed to work well empirically in this study, future work involves developing more sensitive and statistically grounded methods for identifying potential confounders.”

- Response to Weakness 3 in *Evaluation of Potential Confounders*: “Each center can have different patient populations and treatment patterns that shift the only structured adjusted survival rates. For instance, at our center we have a busy high-dose-rate brachytherapy program which is an attractive option for fit patients with few comorbidities who might otherwise receive surgery. This would be expected to bias the survival outcomes in favor of radiation, as observed in our study. We seek to uncover potential confounders from the text that can reduce bias when performing retrospective studies, whichever way the bias lies.”

Regarding PSA scores, Most of the large prostate cancer registry studies we mention in the Introduction also did not include PSA as a predictor, so this limitation is not confined to our study. For instance, study using the SEER database, Abdollah et al 2012 states: "PSA was not available in most cases. Thus, we could not stratify patients by PSA. However, the latter is strongly related to tumor stage and grade, which were considered in our analysis." We do include tumor grade in the structured data, and there is a strong correlation between tumor grade and PSA (Babaian & Smith 1991). We have included a shortened version of this into the Discussion (as seen above).

We were also prevented from running additional experiments for PSA identification because the Stanford system on which we perform high-risk data analysis has been unavailable since January due to system-wide failure.

Abdollah, F., Schmitges, J., Sun, M., Jeldres, C., Tian, Z., Briganti, A., ... & Karakiewicz, P. I. (2012). Comparison of mortality outcomes after radical prostatectomy versus radiotherapy in patients with localized prostate cancer: a population-based analysis. *International Journal of Urology*, 19(9), 836-844.

Babaian, R. J., & Smith, D. B. (1991). Effect of ileal conduit on patients' activities following radical cystectomy. *Urology*, 37(1), 33-35.

Reviewer #4 (Remarks to the Author):

The authors have conducted a very interesting and thorough study in the confines of the many limitations of non-randomized data, the problems of EMR records, and the large selection bias present in the tested comparisons in prostate and lung cancer. The authors have done a great job addressing the reviewer comments.

Thank you!

Reviewers' Comments:

Reviewer #1:

Remarks to the Author:

The authors have addressed my concerns to properly calibrate the contributions of this work.

Reviewer #3:

Remarks to the Author:

Nothing further.